# Intercomparison of IAGOS-CORE, IAGOS-CARIBIC and WMO/GAW-WCCOS Ozone Instruments at the Environmental Simulation Facility at Jülich, Germany

Herman G.J. Smit[1], Torben Galle[1,4], Romain Blot[2], Florian Obersteiner[3], Philippe Nédélec[2], Andreas Zahn[3], Jean-Marc Cousin[2], Ulrich Bundke[1], Andreas Petzold[1], Valerie Thouret[2], Hannah Clark[2]

[1] Institute of Climate and Energy Systems: Troposphere (ICE-3), Forschungszentrum Jülich (FZJ), Jülich, Germany.
[2] Laboratoire d'Aerologie, Université Toulouse III - Paul Sabatier, CNRS, Toulouse, France.
[3] Institute of Meteorology and Climate Research (IMK-ASF), Karlsruher Institut für Technologie (KIT), Karlsruhe, Germany.
[4] Previously published under the name Torben Blomel.

*Correspondence to*: Herman G.J. Smit (h.smit@fz-juelich.de)

**Abstract.** As part of the Quality Assurance (QA) plan of the In-service Aircraft for a Global Observation System (IAGOS), IAGOS-CORE and IAGOS-CARIBIC UV-photometer instruments have been compared with the dual-beam UV- Ozone ($O_3$) PhotoMeter (OPM) of the World Calibration Center of Ozone Sondes (WCCOS) at the Forschungszentrum Jülich in an environmental simulation chamber. The WCCOS was established over 30 years ago as part of the WMO-GAW measurement quality program of the global ozonesonde network, in which the OPM instrument serves as the ozone reference standard. In the simulation chamber, pressure, temperature, and ozone concentration can be controlled at quasi-realistic flight conditions between the Earth surface (~1000 hPa) and ~35 km altitude (5 hPa). During the intercomparison, different ascent/descent and cruise altitude profiles of ozone, pressure and temperature have been simulated between the surface and ~12.5 km altitude (200 hPa).

In general, the two $O_3$ instruments P1-O3 (IAGOS-CORE) and CAR-O3 (IAGOS-CARIBIC) showed good agreement with the OPM reference standard within 5-6 %. At a pressure of 400-500 hPa the agreement was even within 2 %. The observed differences are small but systematic and reproducible during this experiment. CAR-O3 showed a small, pressure-independent deviation of -2 ± 1.5 % compared to the OPM. P1-O3 revealed $O_3$ deviation to the OPM which changes with pressure of about +2% at 1000 hPa to -3% at 400 hPa, which might be an artefact on the experimental set-up and subject for further investigations. This intercomparison is a first step of the long-term goal to make the global ozone sonde data (GAW-NDACC-SHADOZ-GRUAN) and IAGOS-O3 (CORE: P1-O3, CARIBIC: CAR-O3) data traceable to one common reference, the OPM instrument of WCCOS. Recommendations are made for further regular (every two to three years) intercomparison of the operational instruments to ensure external consistency in general and specifically towards the synergy of IAGOS-O3 and ozonesonde data. An important gap in such intercomparison studies is the lack of a reference ozone instrument operated at reduced pressures at any National Metrological Institute in the world. For observation networks measuring vertical ozone profiles, it is essential to close this gap to enable the traceability of ozone measurements from different platforms to one reference standard. This is crucial to harmonize long-term ozone records to detect any changes of ozone in the free atmosphere.

## 1. Introduction

Ozone ($O_3$) is both chemically and radiatively one of the most important trace gases in the atmosphere. It forms the stratospheric ozone layer shielding the Earth's surface from harmful UV sunlight (WMO/UNEP, 2023) and is the major precursor of the hydroxyl radical (OH), the principal chemical detergent controlling the oxidation capacity (e.g. Thompson et al., 1992) and air quality in the troposphere (e.g. Cooper et al., 2014). Tropospheric ozone is also a potent natural and anthropogenically influenced greenhouse gas (IPCC, 2023). Monitoring the vertical ozone distribution on a regional as well as a global scale is essential for understanding long-term changes in both tropospheric and stratospheric ozone, as each may be affected by changes in the dynamics or chemistry of the atmosphere.

Besides the traditional balloon borne ozonesonde network (Smit et al., 2021) to sample tropospheric ozone, in the 1990's new ozone measuring platforms started their routine operations such as Lidar (e.g. McDermid et al., 1991; Ancellet et al., 1998), FTIR (e.g. Schneider et al., 2005; Vigoroux et al., 2008) and the in-service aircraft programs of MOZAIC (Measurement of OZone and water vapor by Airbus In-service airCraft) (Marenco et al., 1998a) and CARIBIC (Civil Aircraft for the Regular Investigation of the atmosphere based on an Instrumented Container) (Brenninkmeijer et al., 1999) . Both in-service aircraft programs have been joined since 2011 into the IAGOS (In-service Aircraft in a Global Observing System) long-term monitoring programme (https://www.iagos.org; Petzold et al., 2015) as part of the European Research Infrastructure for global observations of atmospheric composition (Petzold et al., 2024). During normal scheduled flight operation, IAGOS measures in situ ozone mixing ratios at cruise altitude (10-12.5 km) and provides vertical profiles of ozone from the surface to cruise altitude during take-off and landing. Since August 1994 and over more than 70,000 flights are archived in the IAGOS-database (https://iagos.aeris-data.fr). The data are widely used for climatological and trends analysis (e.g. Petetin et al., 2016; Cohen et al., 2018; Gaudel et al., 2020; Wang et al., 2022; Van Malderen et al., 2025) as well as for model evaluations (e.g. Hu et al., 2017; Wagner et al., 2021).

Crucial for such long-term observations is to prove and monitor their long-term stability as well as the traceability of the instruments to a reference instrument on a regular basis. This can be done by checking the flown instruments on their internal and external consistency. The internal consistency of the IAGOS ozone instruments and their long-term measurements have been evaluated by Blot et al. (2021) and regular procedures have been developed to ensure the internal consistency over time. External consistency checks have been done in the past through in-flight comparison with ozonesonde measurements within a certain coincidence of space and time (Thouret et al., 1998; Staufer et al., 2013, 2014; Tanimoto et al., 2015; Tarasick et al. 2019; Wang et al., 2024). Over more than 25 years of observations, good agreement (within 5-10%) has been achieved between the observing platforms, although ozonesondes tend to consistently measure about 5% more than the aircraft instruments do.

In this study, the external consistency of the IAGOS (CORE and CARIBIC) ozone UV photometer instruments has been investigated through intercomparison with the ozone photometer (OPM) of the World Calibration Centre of Ozone Sondes (WCCOS, https://www.wccos-josie.org/en) at the Forschungszentrum Jülich (FZJ) at their environmental simulation facility to calibrate airborne ozone and water vapor sensors. The WCCOS is established as part of the WMO-GAW measurement quality assurance plan of the global ozonesonde network, whereby the OPM instrument serves as the ozone reference instrument. In the GAW-WCCOS simulation chamber, pressure, temperature, and ozone concentration can be controlled to produce quasi-realistic atmospheric conditions between 1000 hPa (surface) and 5 hPa (upper stratosphere) (Smit et al., 2000). The IAGOS-CORE $O_3$ instrument (here called "P1-O3") is part of the so-called IAGOS-CORE package P1 that is approved as EASA certified aeronautical equipment. Several Package P1 units are operated on commercial

Airbus A340 and A330 aircraft (in 2024: 14 P1-units on 10 different aircraft of 8 international airlines). $O_3$ volume mixing ratio (VMR) measurements are performed for every flight from take-off to landing, including cruise legs at about 180-250 hPa. The tested CARIBIC instrument (here called "CAR-O3") is part of the CARIBIC container laboratory and flown since 2010 on board an Airbus A340 by Lufthansa (Brenninkmeijer et al., 2007). This intercomparison is a first step of the long-term goal to get the global ozone sonde data (GAW-NDACC-SHADOZ-GRUAN) and IAGOS-$O_3$ (CORE: P1-O3 & CARIBIC: CAR-O3) data traceable to one common reference (i.e. OPM of WCCOS).

The key objective of the intercomparison is to investigate the performance of the three ozone UV-photometer instruments (OPM, P1-O3, CAR-O3) under controlled laboratory conditions in the ESC, thereby, simulating typical flight conditions of atmospheric pressure, temperature and ozone concentration between the surface and cruise altitude (Z=10-12.5 km). During the intercomparison different ascent/descent and cruise altitude profiles of ozone have been simulated. This paper presents and discusses the major results of the observed performance of the different instruments in quantitative terms. An outlook will be given on how to have ozone measurements of IAGOS and ozonesondes both traceable to one common ozone reference instrument, i.e. the OPM of the WCCOS chamber.

## 2. Experimental Details

### 2.1 Ozone UV-Photometer Instruments of IAGOS and WCCOS

The principle of the three UV-ozone photometer instruments involved in the intercomparison is based on the spectroscopic UV-absorption measurement of ozone at a wavelength around 254 nm at a well-defined sample path length according to Beer-Lambert absorption law:

$$Ln\left(\frac{I_t}{I_0}\right) = -L \cdot \sigma_{O3} \cdot C_{O3} \tag{1}$$

where $I_o$ (= zero mode) and $I_t$ (=sample mode) are the lamp intensities at the detector when the absorption cell contains the sampled gas with and without removal of the ozone, respectively. $L$ is the length of the absorption cell, $\sigma_{O3}$ is the molecular absorption cross section of ozone at a wavelength of about 254 nm, and $C_{O3}$ is the average concentration of ozone in the absorption cell. Since $L$ and $\sigma_{O3}$ are well known quantities, and the transmittance $R_t = I_t/I_o$ of the absorption cell is determined by the ratio of the two observed signal intensities of the photo detectors in sample and zero mode, respectively, then the ozone concentration $C_{O3}$ can be derived (Eq.1). Through additional measurement of the pressure $P_C$ and temperature $T_C$ inside the absorption cell the volume mixing ratio of ozone $\mu_{O3}$ can be derived from $C_{O3}$.

$$\mu_{O3} = -\frac{k}{L \cdot \sigma_{O3}} \cdot \frac{T_C}{P_C} \cdot Ln\left(\frac{I_t}{I_0}\right) \tag{2}$$

where $k$ is the Boltzmann constant

All instruments use the same widely applied UV-absorption cross section $(\sigma_{O3}= (1{,}147 \pm 0.024) \times 10^{-17} \, cm^2 \, molecule^{-1})$ determined by Hearn (1961). In 2025 a new cross section $(\sigma_{O3}= (1{,}1329 \pm 0.0035) \times 10^{-17} \, cm^2 \, molecule^{-1}$: CCQM.O3.2019 ( https://www.bipm.org/en/gas-metrology/ozone), by Hodges et al., 2019) will be introduced in the global ozone ground-

based monitoring networks (CCQM-GAWG, 2024) which is about 1.29 % lower, however, this will have no impact on the
results of the present intercomparison.
All three ozone instruments are dual-beam UV-photometers that have two identical UV-absorption cells, each alternating
between reference mode (ozone-free air generated by directing it through an ozone scrubber being $CuO/MnO_2$) and sample
mode. A valve assembly alternates the scrubbed air between the two cells, such that one cell is in null mode while the other
cell is in sample mode or vice versa. The mode alternation compensates for changes in the light transmission through the
absorption cells (e.g. due to temperature driven mechanical changes or changes of the reflectivity of the cells due to
changing surface coatings) and finally doubles the measurement frequency. Although the principle of operation is similar
for all three photometer types, the instrumental layouts have significant differences. Specifications of the P1-O3, CAR-
O3 and OPM ozone UV-photometer instruments participating in the intercomparison are summarised in Table 1. In
general, the overall instrumental relative uncertainty is predominantly determined by the uncertainty of $\sigma_{O3}$, the molecular
absorption cross section of ozone. For in situ measurements, sampling uncertainty must be considered. This uncertainty
depends on the design of the air sampling (use of pump in inlet line or not), the use of proper material (e.g. PTFE) to avoid
ozone losses at the walls, and the thermal concept and the electronic design. Therefore, regular pre- and post-flight tests
and characterization of the instruments are essential.

### 126 2.1.1 GAW-WCCOS Ozone Photometer (OPM)

The dual-beam UV-absorption ozone photometer (OPM) of the WCCOS serves as the reference instrument. It was
developed by Proffitt and McLaughin (1983) for use on stratospheric balloons. A low pressure Hg-lamp serves as a UV
light source. The overall uncertainty is ±2 % at P=1000-10 hPa. The instrument serves as reference (standard) of the GAW
global ozonesonde network. The OPM is enclosed in a Styrofoam box, mounted inside a cylindrical vacuum tank which is
connected to the simulation chamber and thus operates at the same pressure level as inside the simulation chamber. Details
of the instrument and the data processing, including uncertainty budget are described in Proffitt and McLaughin (1983).
No ozone reference instrument running at reduced pressures exists at any NMI (National Metrological Institute) in the
world. This means that before and after the intercomparison, the OPM could only be compared at laboratory pressure
conditions (1000 hPa) with a commercial, NIST-traceable "surface" ozone UV photometer of Thermo Electron Instruments
(Model TEI-49) at volume mixing ratios between 0 and 200 ppbv. The agreement was within ±1 ppbv below 100 ppbv
and ±1% above. No systematic bias was observed. Validation of the performance of the OPM at reduced pressures could
only be done based on the evaluation of the measured physical parameters of the OPM as described in Proffitt and
McLaughin (1983).

### 140 2.1.2 IAGOS-CORE Ozone Instrument (P1-O3)

The ozone monitor P1-O3 in IAGOS-CORE is a modified Thermo Scientific (Model 49i) dual beam UV-photometer
integrated together with a CO-infrared monitor in a special aeronautic flight box (Nédélec et al., 2015). The P1-O3
monitor measures ozone at cabin air pressure conditions. Hereby, one UV-absorption cell is in measuring-mode and the
second cell is in zero-mode. In zero mode the ozone is removed from the sampled air by an ozone scrubber ($MnO_2$-
catalyst filter) before the air sample enters the cell that is in zero-mode. Alternating, every 4 seconds (3 s for air flushing
the cells and 1s for the measurement), the cells are switched from sample into zero-mode and vice versa. The pressure
and temperature in the absorption cells are measured to derive the ozone volume mixing ratio from the measured amount
of light absorbed by ozone using Beer's absorption law.  Similar as in the OPM a low-pressure Hg-discharge lamp serves
as UV-light source.
During flight, ambient air is sampled through a forward-facing pitot tube and compressed to cabin air pressure by the
pump (Thomas 118ZC20/24) located in the Pump Box and then fed into the manifold at the inlet of P1-O3, dividing the
total airflow into the nominal sample flow of 4 vol. l/min for the O3 and CO monitors and an excess flow, respectively.
Thereby, the excess air flow is continuously monitored to ensure that the minimum required volume-flow of Pump Box
(25 vol-l/min at ground, 5 vol-l/min at cruise altitude) is obtained. To prevent ozone losses due to physical and chemical
interactions on the walls of the sampling lines, the pitot inlet tube and the interior of the pumps of the Pump Box are
coated with PTFE and all tubing are made of PTFE too.
Before and after aircraft operation (or each ~6 months, respectively), each P1-O$_3$-instrument is checked (without the
Pump Box) against a Thermo Scientific model 49PS reference instrument at several concentration levels to evaluate the
instrument is responding linearly to ozone within 1 %.  In addition, each P1-O3 instrument is sent once a year to the
French Laboratoire National d'Essais (LNE) for comparison with an instrument with measurements traceable to the
National Institute of Standards and Technology (NIST). The overall uncertainty is better than $\pm$ 2 ppbv for O$_3$ < 100 ppbv
and $\pm$2 % for O$_3 \geq$ 100 ppbv. (Nédélec et al., 2015).
Each flown P1-Package (P1-O3 plus Pump Box) for IAGOS-CORE is compared before and after flight periods with the
same MOZAIC-aircraft system that is used to check the performance of the flown packages since the beginning of the
program (Marenco et al., 1998).  Therefore, it is possible to remove systematic biases in the long-term time series and the
resulting measurement uncertainty should represent only the contribution from random errors (Blot et al, 2021). More
details of Package IAGOS-P1 (Pump Box and P1-O3 instrument) and its operation are given by Nédélec et al. (2015) and
Blot et al. (2021).

### 169 2.1.3.  IAGOS-CARIBIC Ozone Instrument (CAR-O3)

The IAGOS-CARIBIC (CAR-O3) UV-photometer ozone instrument is fully custom-made and likewise applies a dual
beam configuration. In zero-mode the ozone is removed using a MnO$_2$-scrubber controlled at 38°C for maximum
efficiency of 100%. Two three-way valves toggle each 4s to guide sample air and zero air alternatively between the two
absorption cells. Each measurement takes 2 s and is preceded by flushing the cells for 2 s.
In contrast to commercial ozone monitors, the instrument uses a UV-LED (Seti, TUD59H1B) as a light source (see
section 2.1 of Zahn, 2016). The LED light is guided into the two sample cells (to ~47% each) using a beam-splitter. The
remaining 6 % is measured by the opposing reference diode to actively control the LED (further stabilized at 20°C using
a Peltier-element) to constant light emission with an uncertainty of $\sim 10^{-4}$ (which is not possible with the conventionally
used low-pressure Hg discharge lamps).  However, since the UV-LED emission spectrum has a full-width half-mean
(FWHM) of ~11 nm and may age, it is regularly (about every 3 months) calibrated against a reference UV photometer
using a low-pressure Hg discharge lamp as UV light source and applying the UV absorption cross section of Hearn
181 (1961).

Two photodiodes (Hamamatsu S1226) at the end of the cells measure the UV light intensity using a two-channel (not
multiplexing) 24-bit sigma-delta amplifier. Temperature is measured on the outside and the inside of the cells. Pressure is
measured directly at the exhaust of the cells. Sample flow during aircraft operation of CAR-O3 is determined by the ram-

pressure through the CARIBIC inlet system. This guarantees a minimum flow of 1.5 vol-l/min at cruise altitude. During the experiments reported here (without the ram-pressure on aircraft), a flow of ~2 vol-l/min was enforced by a pump downstream of the instrument in combination with a needle valve for manual control of the flow. The main specifications are listed in Table 1. Further details of handling and data processing are described in Zahn (2016) as well as Obersteiner (2024, https://doi.org/10.5281/zenodo.11104076).

The measured precision (1-sigma) is 0.06 ppb at 1000 hPa and the response time of 4s. A simple calculation based on the photon flux reaching the photodiodes (inferred from its photosensitivity and the measured photo current) and the detected photo current noise indicate that this precision exactly agrees (to within 10-15 %) with the measured shot noise, that is, CAR-O3 is quantum-noise limited and higher precision can only be reached with a stronger UV-LED or a longer absorption path length.

The total uncertainty of 2 ppb or 2% (whatever is higher) is dominated by the uncertainty of the $O_3$ cross section around 255 nm (Zahn, 2016). After the initial calibration the CAR-O3 instrument is regularly (typically each 3-4 months) compared in the laboratory with a reference photometer and once a year with a 2.7 m long-path UV photometer (by UMEG).

**Table 1.** **Specifications of the P1-O3, CAR-O3 and OPM ozone UV-photometer instruments participating in the intercomparison.**

| Property | P1-O3 IAGOS-CORE | CAR-O3 IAGOS-CARIBIC | OPM GAW-WCCOS |
|---|---|---|---|
| **Light Source** | Hg-Lamp (254 nm) | UV-LED (near 254 nm) | Hg-Lamp (254 nm) |
| **UV Abs. Length** | 38 cm | 38 cm | 40 cm |
| **Pressure** | Cabin | Ambient | Ambient |
| **Inlet** | Pitot (forward) + Compressor | Pitot (forward) + ram-pressure | N/A |
| **Sample Volume Flowrate** | 24 lv/min ( 2 lv/min for P1-O3 & 2 lv/min for P1-CO ) | 2 lv/min | 8 lv/min |
| **Response Time** | 4 seconds | 4 seconds | 2 seconds |
| **Precision** < 100 ppbv ≥ 100 ppbv | ± 1 ppbv ± 1 % | ± 0.1 ppbv ± 0.1 % | ± 1 ppbv ± 1 % |
| **Overall Uncertainty** < 100 ppbv ≥ 100 ppbv | ± 2 ppbv ± 2 % | ± 2 ppbv ± 2 % | ± 2 ppbv ± 2 % |
| **Reference** | Nédélec et al., 2015 | Zahn, 2016 | Proffitt and McLaughlin, 1983 |

203

## 2.2. Environmental Simulation Facility: GAW-WCCOS

### 2.2.1 GAW-WCCOS Simulation Chamber

The GAW-WCCOS simulation chamber established at the Forschungszentrum Jülich (FZJ) is designed to investigate the performance of different types of balloon-borne ozone sensors as well as airborne humidity sensors to measure the vertical distribution of atmospheric ozone and water vapor, respectively (Smit et al., 2000). The key component of the facility is a simulation chamber with a test room volume of about 500 liters (80x80x80 cm) whose pressure as well as temperature can be dynamically regulated between 5 and 1000 hPa and between 200 and 300 K at temperature rates between -2K/min and +5K/min. The volume mixing ratio of ozone can be dynamically regulated between 5 and 10000 ppbv to simulate typical atmospheric ozone levels between the surface and 35 km altitude. Since 1994, the facility has been established as the World Calibration Centre for Ozone Sondes (WCCOS) as part of the QA/QC-management plan of the Global Atmosphere Watch (GAW) program of the World Meteorological Organization (WMO). In the scope of this framework since 1996, international JOSIE (Jülich Ozone Sonde Intercomparison Experiment) campaigns have been conducted to assess the performance of the major types of ozone sondes used within the global network of ozone sounding stations (Smit et al., 2007, 2024; Thompson et al., 2019). The dual beam UV-photometer OPM (section 2.1.1) serves thereby as an ozone reference. The entire simulation process is automated by computer control to guarantee reproducible ambient conditions. JOSIE observations have demonstrated that the experimental set-up of the WCCOS simulation chamber experiment has a reproducibility of about ±1%. Details of the facility and its use as WCCOS are given by Smit et al. (2000).

### 2.2.2 Ozone Profile Simulator (OPS)

The Ozone Profile Simulator (OPS) unit of GAW-WCCOS (Smit et al., 2000) is used to simulate reproducible pressure dependent ozone profiles dynamically in time. Therefore, a separate gas mixing system is installed to supply equivalent air samples to up to four ozone sensors plus the UV-photometer (OPM) with pre-set ozone concentrations. Ozone is photolytically generated by UV-irradiation in a zero-grade airflow through a quartz glass (Suprasil) cuvette using a low-pressure Hg-discharge lamp. Via the photodissociation of oxygen molecules at a wavelength of 185 nm, ozone is produced at high levels of 0.1-0.2 % in a constant flow of 50 cm3/min through the quartz glass cuvette (pressurized at 4000 hPa, volume: ~40 $cm^3$). To vary the ozone volume mixing ratio between 10 and 10000 ppbv at different air pressures, the high-ozone airflow is dynamically diluted by a two-staged mixing with zero-grade air flows. All air flows are regulated by mass-flow controllers (Smit et al., 2000). The air used is dried and purified such that any sensitivities of the UV-Photometers to humidity or sudden changes of it (Wilson and Birks, 2006) can be excluded. The sample flow is connected to a glass manifold inside the simulation chamber to feed the different $O_3$-instruments, whereby excess air can flow via an exhaust, such that the inlet tubes of all connected instruments are at the same pressure condition as inside the WCCOS-chamber.

## 2.3 Experimental Design Intercomparison

### 2.3.1 Experimental Setup

The schematic of the experimental setup is shown in Figure 1. Ozone-containing air is produced in the OPS and fed into a gas manifold located inside the simulation chamber. The inlet tubes of the three ozone instruments are connected to the manifold via gas-feed through (all made of PTFE). CAR-O3 uses a 2 m tube (ID = 4 mm), while the P1-O3 the inlet line goes via the P1-Pump Box that compresses the sample air to cabin or (here) laboratory pressure before entering the P1-O3

240 instrument. The OPM, mounted in a vacuum tank connected to the simulation chamber, is at the same pressure condition
241 as inside the chamber.

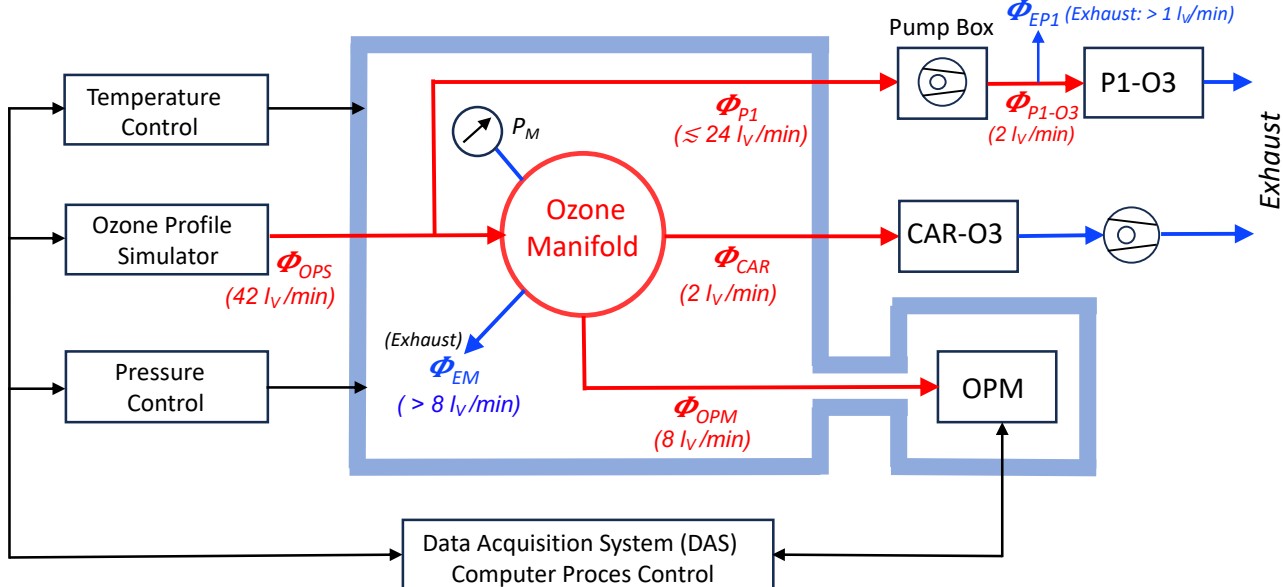

**Figure 1:** Schematics of the experimental setup for the intercomparison at the WCCOS, showing the ESC with OPM of the WCCOS, the connection to the IAGOS-CORE and IAGOS-CARIBIC ozone instruments, the ozone manifold located inside the simulation chamber and its control systems, including the computer-controlled DAS.

The sample manifold consists of a spherical glass vessel with a volume of about 150 cm$^3$ with radially arranged connections
to the individual ozone instruments with the inlet of the simulated ozone air flow $\Phi_{OPS}$ being in the centre of the manifold.
Excess air is exhausted via an additional tube such that the manifold is kept to the sample volume pressure (measured by
a pressure sensor) and to prevent the inlet lines of the ozone instruments from overpressure effects that may cause
measurement artefacts.
For the JOSIE experiments (for testing ozone sondes), the volume flow rate of the simulated ozone air flow $\Phi_{OPS}$ is kept
constant at 12 vol-l/min which is sufficient to provide four ozone sondes (maximum 4 x 0.25 vol-l/min) and the OPM
(maximum 8 vol-l /min). For the IAGOS-ozone intercomparison higher flow rates were required, sufficient to supply all
instruments (see Sample Volume Flowrate in Table 1). The total volume flow rate is at least 36 vol-l/min. To ensure a
significant exhaust flow at the manifold, we thus increased the typical volume flow of 12 vol-l/min by an additional 30
vol-l/min flow controller to obtain a total volume flow $\Phi_{OPS}$ of 42 vol-l/min and thus an exhaust flow of the manifold of 6
vol-l/min (Fig.1). The pressure $P_M$ inside the manifold was monitored to ensure to keep it a few hPa higher than the
pressure in the simulation chamber itself to avoid any leakage effects of air from the chamber into the manifold. The P1-
O3 sample flow we had to branch off from the ozone profile simulator flow before entering the manifold (Fig.1), because
it was shown that the high sampling volume rate of P1-O3 pump box would otherwise cause leakage effects when P1-O3
was directly connected with a Teflon fitting at the inlet glass tube of the manifold.
**2.3.2 Simulation of Realistic Flight Conditions**
It is essential to operate the chamber at appropriate pressure conditions to simulate realistic flight conditions that the IAGOS
instruments experience when connected to the air inlets. Both air-sample inlets (of IAGOS-CORE and IAGOS-CARIBIC)
are facing forwards and thus use the dynamic (ram) air pressure generated by the high speed of the aircraft. On IAGOS-
CARIBIC a special inlet configuration hinders (aerosol and cloud) particles larger than ~2 μm to enter the sampling line.
At the maximum cruise altitude of about 12.5 km, the lowest static air pressure is 180 hPa at a typical aircraft speed of
Mach = 0.81+/-0.02 causing an adiabatic compression factor of about 1.6. In an ideal case, this leads to a dynamic rem air
pressure of about 100 hPa.  However, in practice some pressure losses of about 30 hPa have to be considered, such that the
lowest total air pressure inside the inlets is about 250 hPa. Note, however, as P1-O3 runs a pump to compress sampled air
to cabin pressure (here laboratory pressure) before entering P1-O3 instrument, the pressure ranges of P1-O3 and CAR-O3
covered by our tests are different, but for both instruments span the relevant pressure ranges between surface and cruise
altitude.
A.   IAGOS-CORE = P1-O3
The P1-Pump Box supplied with sample air from the forward-facing inlet system compresses the sampled air to cabin air
pressure. The cabin air pressure is prescribed by civil aviation regulations to be above 750 hPa and usually ranges at 800-
850 hPa at cruising altitude. In-flight, the maximum pressure difference between cabin and the inlet of P1-PU thus is 850-
250 = 600 hPa. For the present laboratory intercomparison we thus must cover the pressure range between 1000 hPa and
400 hPa (= 1000 - 600 hPa).
B.   IAGOS-CARIBIC = CAR-O3
The CAR-O3 instrument does not use a pump, and its inlet pressure is the ambient static pressure, plus the ram-pressure
minus some pressure loss in the sampling line (see above), that is, 250 hPa at maximum cruise altitude. To simulate the
ram-pressure effect (exhaust at 180 hPa), during this laboratory intercomparison the CAR-O3 instrument does not use a
pump at the exhaust to force an air flow of about 2 vol-l/min (Fig.1).

**3  Results**
**3.1. Introduction**
Table 2 gives an overview of the simulation experiments performed.  The first day (12 June 2023) was reserved for
installation of the equipment and for a short test run to ensure proper functioning of equipment and data acquisition of the
different instruments. On the second day (13 June 2023) another test of the P1-O3 and CAR-O3 instruments followed by
sampling outside ambient air. The results of these two tests are beyond the scope of this report. The core of the
intercomparison itself took place on 13 until 15 June 2023 with the four simulation experiments, numbered 3 to 7, which
will be presented here in more detail.


**Table 2. List of intercomparison experiments performed during the IAGOS-WCCOS Ozone Intercomparison (IWOI) campaign between 12 and 15 June 2023 at WCCOS (FZJ/IEK-8, Jülich, Germany).**


| Date | Exp. Nr & Sim. Nr | Profile Type | UTC-Time | Pressure (hPa) | Remarks |
|---|---|---|---|---|---|
| Day#1: 12-06-2023 | #1 & 223 | Test | 13:00-15:00 | 1000-300 | Installation and testing equipment |
| Day#2.1: 13-06-2023 | #2 & NAN | Ambient Air (Day2.1_Ambient) | 07:30-09:30 | 1000 | P1-O3 & CAR-O3 & No OPM |
| Day#2.2: 13-06-2023 | #3 & 224 | Ascent-Cruise-Descent (Day2.2_Profile) | 12:00-17:00 | 1000-400-400-1000 | P1-O3 & CAR-O3 & OPM |
| Day#3.1: 14-06-2023 | #4 & 225 | Ascent-Cruise-Descent (Day3.1_Profile) Cruise: O3 Step-Up/Down | 07:30-11:30 | 1000-400-400-1000 | P1-O3 & CAR-O3 & OPM |
| Day#3.2: 14-06-2023 | #5 & 226 | Discrete Pressure Levels (Day3.2_Profile) Total OPS-Flow: 12 vol-l/min | 11:30-14:00 | 1000-400-250 | CAR-O3 & OPM & No P1-O3 |
| Day#3.3: 14-06-2023 | #6 & 226 | Ascent-profile (Day3.3_Profile) Ascent Zero Ozone | 14:00-15:00 | 1000-250 | CAR-O3 & OPM & No P1-O3 |
| Day#4.1: 15-06-2023 | #7 & 228 | Discrete Pressure Levels (Day3.3_Profile) | 07:00-10:30 | 1000-400 | P1-O3 & CAR-O3 & OPM |


### 3.2 Comparison of P1-O3, CAR-O3 and OPM at a pressure of 400-1000 hPa

### 3.2.1 Experiment #3: Ascent-Cruise-Descent

Experiment #3 (numbering, see second column in Table 2) simulates an aircraft doing an "ascent - cruise altitude - descent"
profile of pressure and ozone volume mixing ratio (Figure 2, 4, 5). The lowest pressure of 400 hPa is to simulate the
maximum pressure difference the P1 pump box must achieve between cruise altitude and about 1000 hPa in the laboratory
(see explanation in section 2.3.2.). In the first part of the simulation, during the ascent and the beginning part of the cruise
phase, the ozone level was maintained at 400-500 ppbv to clean the inlet tubes of the OPM, P1-O3 (including P1-Pump
Box) and CAR-O3 instruments. In the second part, the ozone was lowered to about 100 ppbv. The three missing data
intervals of CAR-O$_3$-instrument were caused by a malfunction of its temperature controller of the UV-LED light source
such that the measured O$_3$ values were rejected and not shown in the graph and excluded from further analysis.

The relative differences in % of the $\mu_{O3}$ (VMR) readings of P1-O3 and CAR-O3, respectively, shown in this study are
consequently defined with respect to the $\mu_{O3}$ readings of the OPM-O3 instrument acting as the reference as follows:
$$Rel. Difference\ of\ P1O3\ \ = \frac{(\mu_{O3,P1O3} - \mu_{O3,OPMO3})}{\mu_{O3,OPMO3}} \qquad (3)$$
$$Rel. Difference\ of\ CARO3 = \frac{(\mu_{O3,CARO3} - \mu_{O3,OPMO3})}{\mu_{O3,OPMO3}} \qquad (4)$$

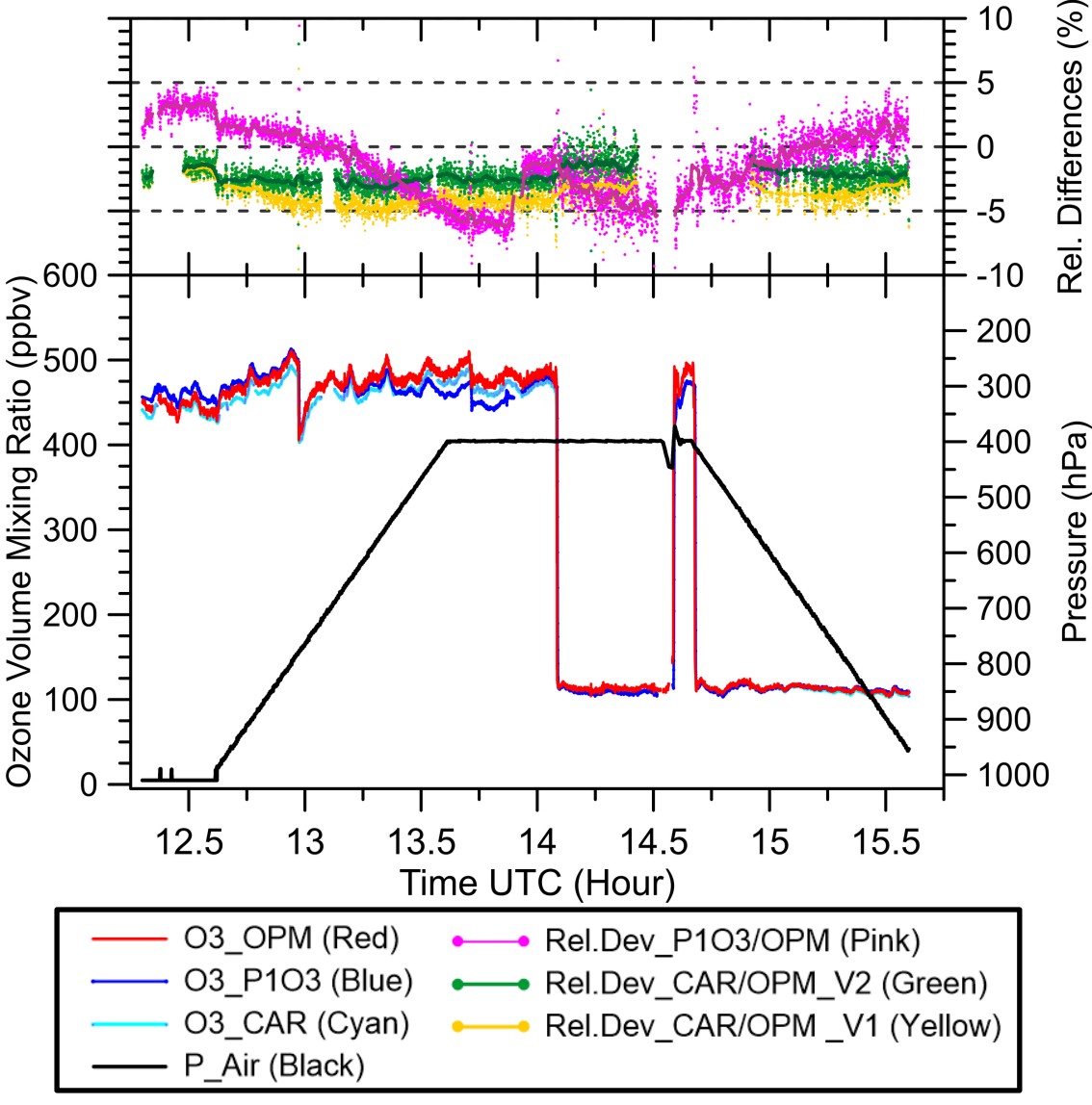

**Figure 2: Experiment #3: Time-series of pressure (black) and ozone volume mixing ratio to simulate an ascent-cruise-descent**
**track of an IAGOS aircraft for P1-O3 (blue), CAR-O3 (cyan=light blue) and OPM (red). The relative differences compared to**
**each other are P1-O3 to OPM (pink) and CAR-O3 (original: V1) to OPM (yellow) and CAR-O3 (pressure-sensor corrected: V2,**
**see text) to OPM (green). Fat solid lines are 3-minute running averages of the relative differences.**

In general, the three instruments follow the simulated ozone profile well and agree among each other between -5 and +2%
(Fig. 2). P1-O3 compared to the OPM shows a pressure dependence, that is, from +3% at 1000 hPa down to -5 % at cruise
altitude conditions. The CAR-O3 instrument initially exhibited an increasing negative offset relative to the OPM of - 1%
at 1000 hPa (at ~12:30) to -4 % at 800 hPa and lower pressures (Fig. 2: yellow curve).  This somewhat strange behaviour
was subject to further investigations on the underlying cause. In a subsequent test (May 2024), KIT (Karslruher Institut für
Technologie), responsible for the operation of the CAR-O3 instrument, found an issue with the electronic analog-digital
converter of the data acquisition card of CAR-O3 that generated a systematic 2.2% difference of the reading of the pressure,
$P_{Cuv}$) inside the absorption cells below a pressure of ~800 hPa (see Figure 3a.). This electronic artefact has been eliminated
and the pressure readings before and after the repair of the AD-converter were compared against an accurate reference

pressure, $P_{ref}$, sensor (Omega HHP360, accuracy: 0.25 hPa). The observed pressure differences, $\Delta P_{Abs} = P_{Cuv.} - P_{Ref.}$ as function of the reference pressure $P_{Ref.}$ (Figure 3a.) are used to correct all original CAR-O3 data (version V1) into the new pressure-sensor corrected CAR-O3 data (version V2). After the repair of the AD-converter, the corrected V2 data show a rather constant, pressure independent, deviation of about -2 % compared to the OPM (Fig. 3b.) We only will present the pressure corrected CAR-O3 data. There exists just one single CAR-O3 instrument that is in flight operation. Meanwhile, all CARIBIC-Ozone data in the IAGOS database (https://iagos.aeris-data.fr/) have been corrected accordingly. For the two similar CAR-O3 type instruments (FAIRO-1 and FAIRO-2) which are flown on the German research aircraft HALO (HALO (High Altitude and Long-Range Research Aircraft) the ADC-cards were configured correctly, such that no correction is needed.

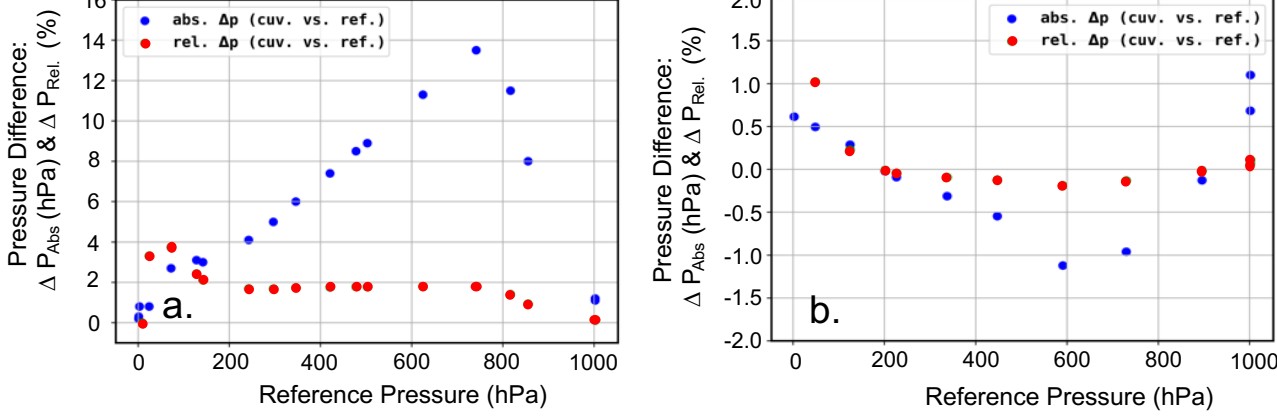

**Figure 3   Comparison of CAR-O3 air pressure sensor, $P_{Cuv}$, inside UV-absorption cell against accurate reference pressure, $P_{Ref,}$ sensor (Omega, HHP360, uncertainty: 0.25 hPa) before (left diagram a.) and after (right diagram b.) solving the electronic artifact of the AD-converter (details see main text). Displayed are the pressure differences $\Delta P_{Abs} = P_{Cuv.} - P_{Ref.}$ in hPa (blue dots) and their relative differences $\Delta P_{Rel.} = \Delta P_{Abs} / P_{Ref.}$ in % (red dots).**

In Figure 4 the identical data (experiment No. 3, see Fig. 2) have been split into the vertical $O_3$-profiles during ascent (Fig. 4a) and descent (Fig. 4b). The behaviour of P1-O3 and CAR-O3 described above occurs identically during ascent and descent, and no indication for any hysteresis effects could be observed. This is also confirmed by the fast responses of both instruments on the sharp upward or downward steps of the simulated ozone levels.

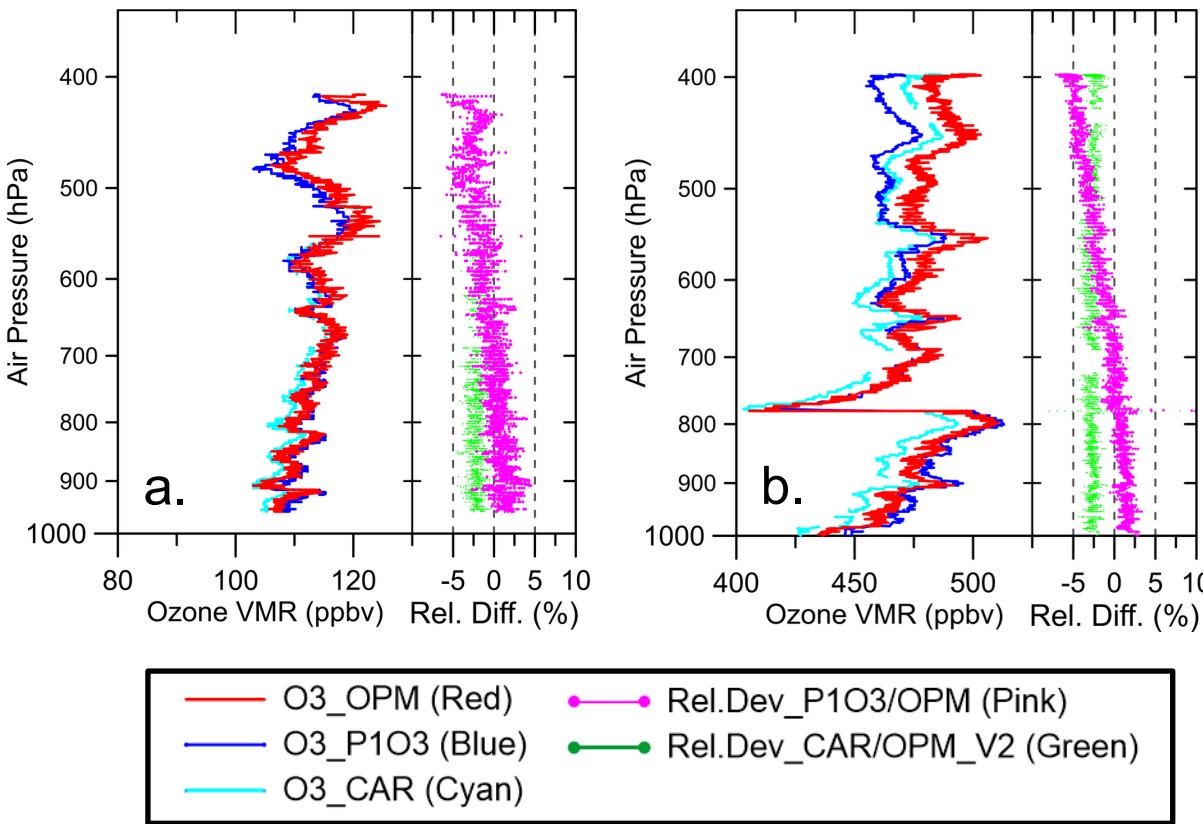

**Figure 4.** Experiment No. 3: Same data (and colours) as in Figure 2 but has been split into ascent (a: left diagram) and descent (b: right diagram). The measured ascent and descent profiles are displayed as ozone versus the simulated pressure ($^{10}$Log scale).

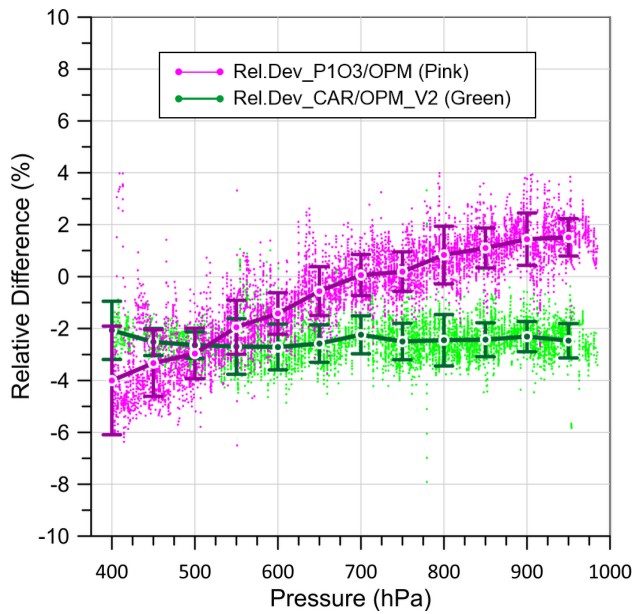

**Figure 5.** Experiment No. 3: Same data (and colours) as in Fig. 2, but relative differences of P1-$O_3$ and CAR-O3 compared to OPM as function of pressure. Thick solid lines are averages over 50 hPa pressure bins with their 1 σ-standard deviation.

The results of this Exp#3 are summarized in Figure 5 that displays the relative differences of P1-O3 and CAR-O3 compared
to the OPM as scatter plot and function of the air pressure inside the chamber. The thick curves are the corresponding
averages over 50 hPa bins with their one standard deviation.

### 3.2.2  Experiment #4: Ascent - Cruise (O₃ steps) - Descent

This simulation experiment is similar to Exp. No. 3, with the following differences:  during ascent and descent the ozone
volume mixing ratio was held at 110 ppbv, while at cruise altitude, the ozone was varied (stepped up and down) at different
levels of 100, 250, 370 and 500 ppbv, see Figures 6 - 8 equivalent to the Figures 2, 4 and 5 respectively.

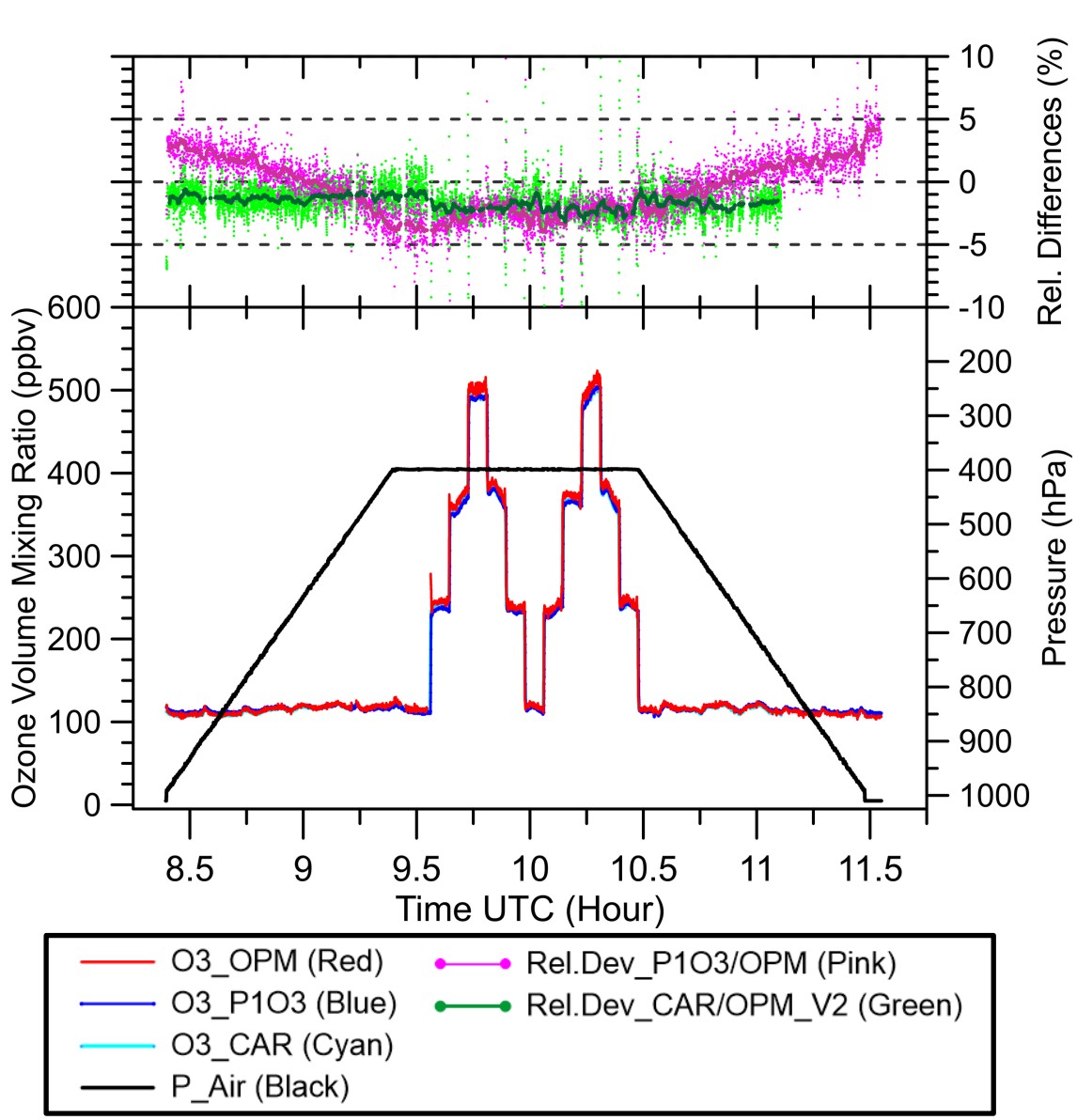

**Figure 6.   Experiment No. 4: Graph and colour coding identical to Fig. 2.**

Also, in this simulation experiment the instruments follow the simulated ozone profile well and agree among each other to
within ± 3%.  From Fig. 7 and 8 it is depicted that the P1-O3 compared to the OPM show a significant decrease with
decreasing pressure, similar as in the previous Exp. No. 3 from +3% at 1000 hPa down to -3 % at 400 hPa (cruise altitude
conditions). The CAR-O3 instrument relative to the OPM revealed a similar behaviour as in Exp. No. 3: - (1.5 – 2) %
deviation that is constant at pressures between 1000 hPa and 400 hPa. Remarkable is that the span and slope of all data are
identical to Exp. #3, but all data are shifted to (0.8 - 1.0) % higher values. Based on this observation we estimate the
reproducibility of the experimental set-up within ± 1%. Further, no indications are found on any memory or hysteresis
effects for both instruments.

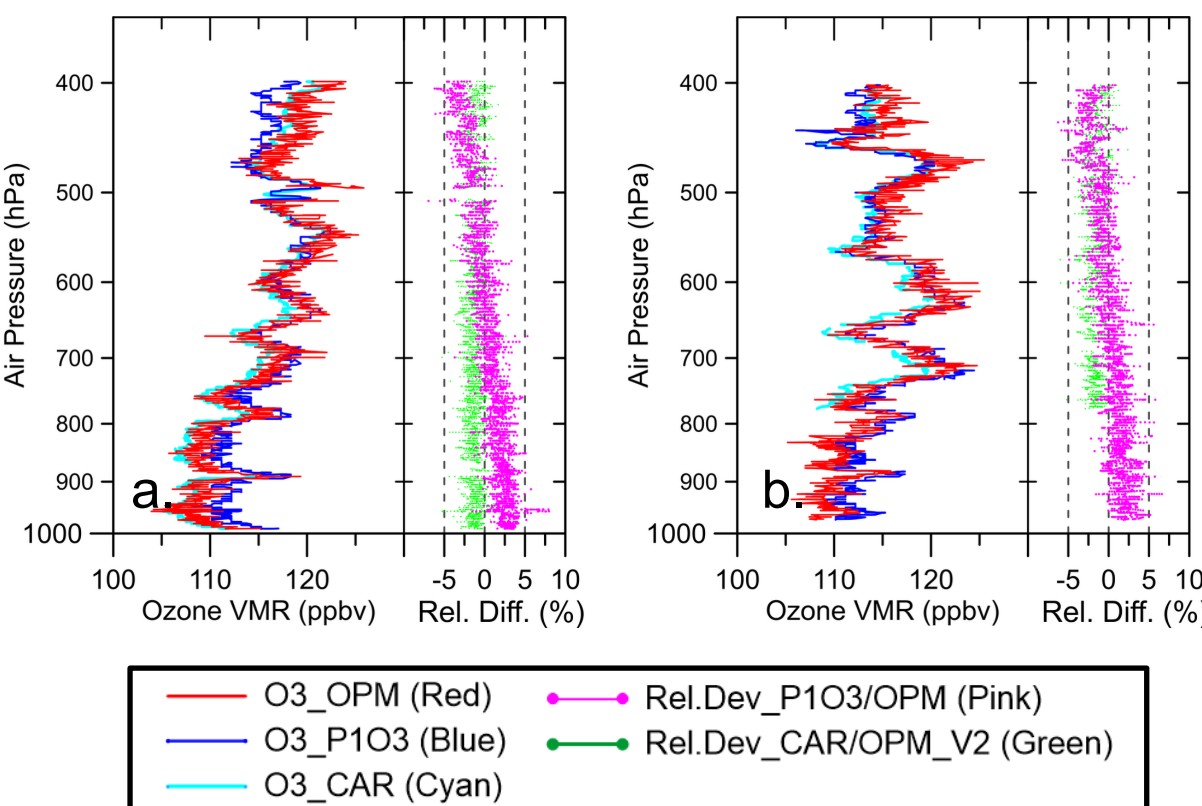

**Figure 7.** **Experiment No. 4: Graphs and colour coding identical to Fig. 4.**

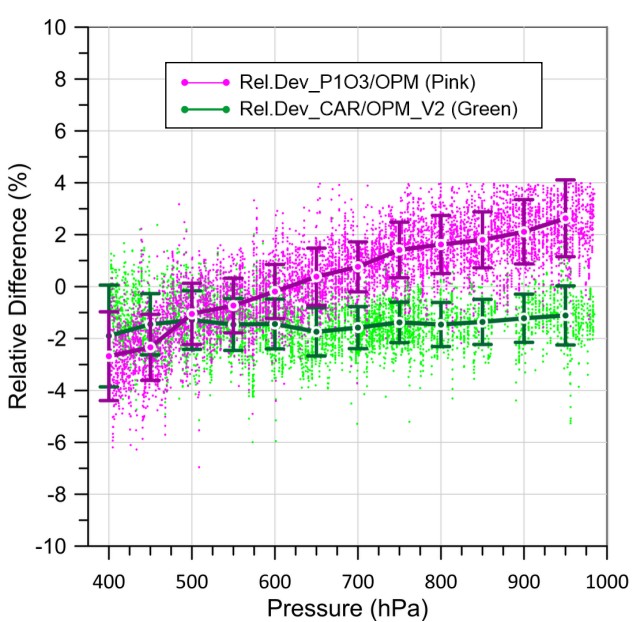

**Figure 8.** **Experiment No 4: Graph and colour coding identical to Fig. 5.**

### 3.2.3. Experiment No. 7: O₃ Step Up/Down at Different Pressure Levels

In this simulation experiment at three different discrete pressure levels (950, 600 and 400 hPa) the ozone levels were varied (step up and down) at discrete values typically found at the corresponding pressure levels, (See Figures 9 and 10).

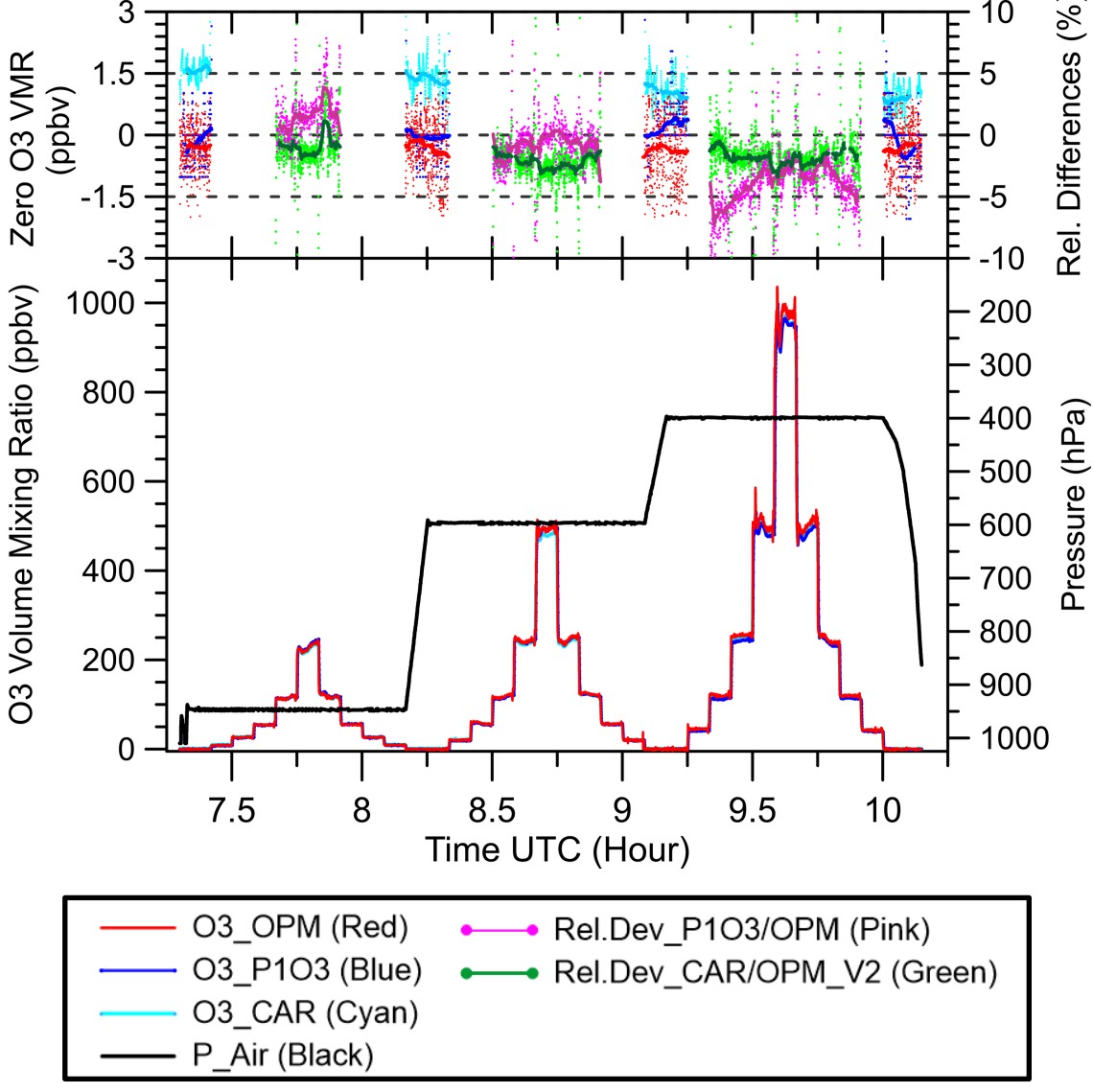

**Figure 9.** Experiment #7: colour coding as listed in Figure 4. In addition, in the upper panel (left vertical coordinate) are plotted the ozone offsets (red: OPM; blue: P1-O3; light blue = cyan: CAR-O3) measured during periods of zero ozone air, whereby fat solid lines are 3-minute running averages of the offsets.

At low pressure around 400 hPa (Fig. 9), P1-O3 shows a small ozone dependent bias to the OPM from -5 % at ~100 ppbv to -2% at ~1000 ppbv. The bias of CAR-O3 relative to OPM is again (as in Exp#3 and #4) around -(1-2) % and is constant over the entire pressure range of 400 - 1000 hPa with ozone volume mixing ratios up to 1000 ppbv. Although the three instruments track changes in ozone levels below 100 ppbv, only the relative differences for the higher ozone levels are shown in Figure 9. This is because at lower ozone concentrations, even small differences between the instruments would result in large relative values. Therefore, to compare the behaviour of P1-O3 and CAR-O3 in more detail, figure 10 shows

ozone VMR scatter plots of P1-O3 versus OPM and CAR-O3 versus OPM, respectively, for the three discrete pressure
levels of 950, 600 and 400 hPa, once for the full ozone ranges upper panel) and once for the lower ozone ranges (lower
panel). The lower ozone VMR levels are more representative of tropospheric conditions (Fig. 10-d, e, f).

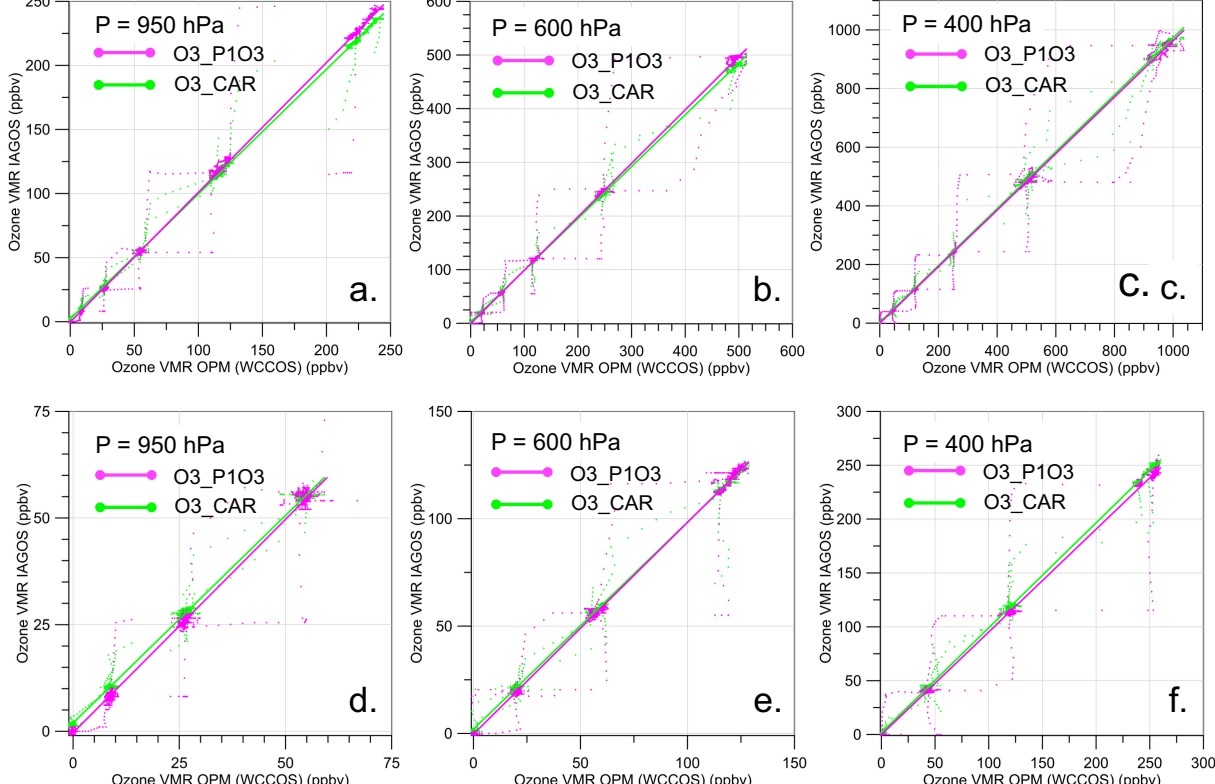

**Figure 10.**    **Experiment #7:**    **Ozone pressures measured by IAGOS instruments versus OPM at different ozone VMR levels**
**(ppbv) for three discrete constant air pressure levels: 950, 600 and 400 hPa. Displayed are the scatter plots of P1-O3 versus**
**OPM (pink) and CAR-O3 versus OPM (green) and the solid straight lines are their linear curve fits. Upper panel displays the**
**full ozone ranges (graphs a., b., c.) and the lower panel the lower ozone ranges (graphs d., e., f.) at the three different air pressure**
**levels.**

The results for each pressure level (950, 600 and 400 hPa) are summarized in Table 3, once for the entire ozone VMR
range and once for the lower ozone VMR level. The offsets of the instruments have been determined in the periods when
measuring zero ozone air by averaging over 5-minute intervals (Fig.9: upper panel, left vertical coordinate). At each
pressure level the slope has been derived from a linear curve fit of the scatter plots of P1-O3 and CAR-O3 versus OPM
(Fig. 10). To investigate any hysteresis effect, the slopes have been determined also for the upward step ozone levels and
downward step levels, the corresponding figures are shown in the supplementary material (Fig. S1 and Fig. S2 for P1-O3
and CAR-O3 against OPM, respectively). All results of slopes and offsets are summarized in Table 3.

From Table 3 it is seen that the behaviour between the three instruments observed at ozone levels larger than about 100
ppbv is consistent with the results obtained from the Exp. #3 and Exp. #4. At lower ozone values below 100 ppbv, however,
the slopes for P1-O3/OPM differ slightly by -(1-2) % compared to their corresponding slopes of P1-O3/OPM derived for
higher ozone values, respectively. Breaking down the slopes into the upward and downward part of the ozone step levels,
P1-O3/OPM reveals a small hysteresis effect of about 2 % which is most pronounced in the lower range of ozone levels.
CAR-O3 shows no hysteresis, neither at the higher nor at the lower ozone levels (Table 3 and Figs. S1 and S2 in the
supplement). The observed differences are not really understood but are still within the experimental reproducibility of
about ±1 % as mentioned in Section 3.2.2.

**Table 3.** **Offsets of OPM, P1-O3 and CAR-O3 determined from zero ozone air measurements (Fig.8) and slope of linear curve**
**fits of scatter plots of P1-O3 and CAR-O3 versus OPM scatter plots (Fig.9), respectively, at three different air pressure levels:**
**950, 600 and 400 hPa for all data (Fig.9), included the corresponding slopes for the upward and downward ozone step levels,**
**respectively (Fig. S1 and S2 in the supplementary material, respectively).**

| Pressure (hPa) | Ozone Range (ppbv) | Ozone Data | P1-O3/OPM Slope | CAR-O3/OPM Slope | OPM Offset at zero $O_3$ (ppbv) | P1-O3 Offset at zero $O_3$ (ppbv) | CAR-O3 Offset at zero $O_3$ (ppbv) |
|---|---|---|---|---|---|---|---|
| 950 | 0 | Zero | -- | -- | -0.25±0.5 | -0.30±0.6 | 1.5±0.2 |
| | 0-250 | All | 1.012 | 0.975 | -- | -- | -- |
| | | Up | 1.007 | 0.973 | -- | -- | -- |
| | | Down | 1.020 | 0.977 | -- | -- | -- |
| | 0-75 | All | 1.002 | 0.973 | -- | -- | -- |
| | | Up Down | 0.980 | 0.971 | -- | -- | -- |
| | | | 1.023 | 0.976 | -- | -- | -- |
| 600 | 0 | Zero | -- | -- | -0.27±0.7 | -0.08±0.3 | 1.5±0.3 |
| | 0-600 | All | 0.995 | 0.972 | -- | -- | -- |
| | | Up | 0.993 | 0.970 | -- | -- | -- |
| | | Down | 1.010 | 0.974 | -- | -- | -- |
| | 0-150 | All | 0.986 | 0.971 | -- | -- | -- |
| | | Up Down | 0.975 | 0.968 | -- | -- | -- |
| | | | 0.990 | 0.974 | -- | -- | -- |
| 400 | 0 | Zero | -- | -- | -0.23±1.0 | -0.32±0.6 | 1.1±0.5 |
| | 0-1100 | All | 0.964 | 0.975 | -- | -- | -- |
| | | Up | 0.958 | 0.970 | -- | -- | -- |
| | | Down | 0.972 | 0.978 | -- | -- | -- |
| | 0-300 | All | 0.954 | 0.974 | -- | -- | -- |
| | | Up Down | 0.949 | 0.971 | -- | -- | -- |
| | | | 0.968 | 0.976 | -- | -- | -- |





### 3.3    Comparison CAR-O3 Versus OPM at 250-1000 hPa Pressure


### 3.3.1    Experiment #5: Discrete Pressure Levels (1000-250 hPa)


To simulate the real cruise altitude conditions for CAR-O3 (see section 2.3.2), a simulation experiment was repeated at
three different pressure levels (1000, 500 and 250 hPa), whereby the ozone volume mixing ratios were kept at levels
between 150 and 250 ppbv. The P1-O3 did not participate in this comparison experiment because the low-pressure level
of 250 hPa is not within the specification of the P1-Pump Box to operate against 1000 hPa laboratory pressure instead of
850 hPa cabin air (i.e., the pressure under real flight conditions, see section 2.3.2). In this simulation the total volume flow
rate of the OPS, $\Phi_{OPS}$ is reduced to 12 vol-l/min. The results are shown in Figure 11.

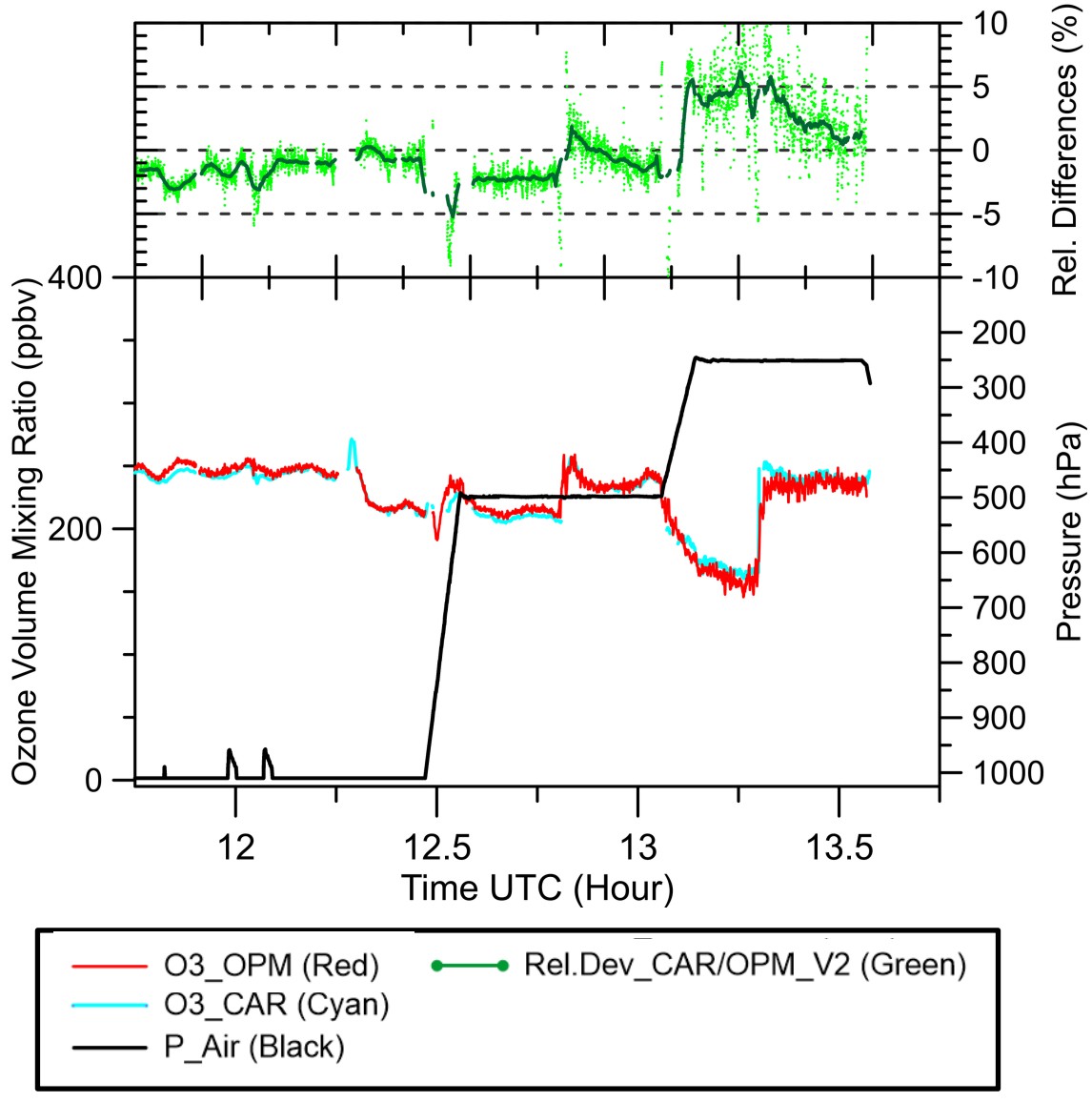


**Figure 11. Experiment #5: Time series of pressure (black) and ozone pressure (mPa) for CAR-O3 (cyan=light blue), OPM**
**(red) and relative deviation of CAR-O3 to OPM (upper panel: green).**

At 1000 hPa and 500 hPa the results are very similar with the results of Exp.#3 and Exp. #4, while at 250 hPa initially CAR-O3 shows slight enhanced values of about + (4-5) % compared to OPM, but after about 10 minutes, the difference declined to + (1-2) %. The cause of this behaviour has been investigated by evaluating the housekeeping data of both instruments (CAR-O3 and OPM) as well as the OPS and ESC; however, no indication of any malfunction of any of the components could be detected. The cause is still not understood; it is a subject for further detailed investigations.

### 3.3.2    Experiment #6: Zero O3 Ascent (1000-180 hPa)

In this experiment the ascent pressure (down to 200 hPa) was simulated while ozone was kept at zero to measure the zero signals of the CAR-O3 and OPM, while P1-O3 did not participate in the experiment. The OPM showed a small negative offset about - (0.02-0.05 mPa), but a rather noisy signal, unrealistic high and most likely due to the enhanced temperatures of the UV-light detector electronics exceeding the 50 ºC threshold that occured during the experiment. The CAR-O3 showed a small positive offset of 0.1 mPa at 1000 hPa that vanishes towards lower pressures, which agrees with results of Exp.#7 (Table 3).

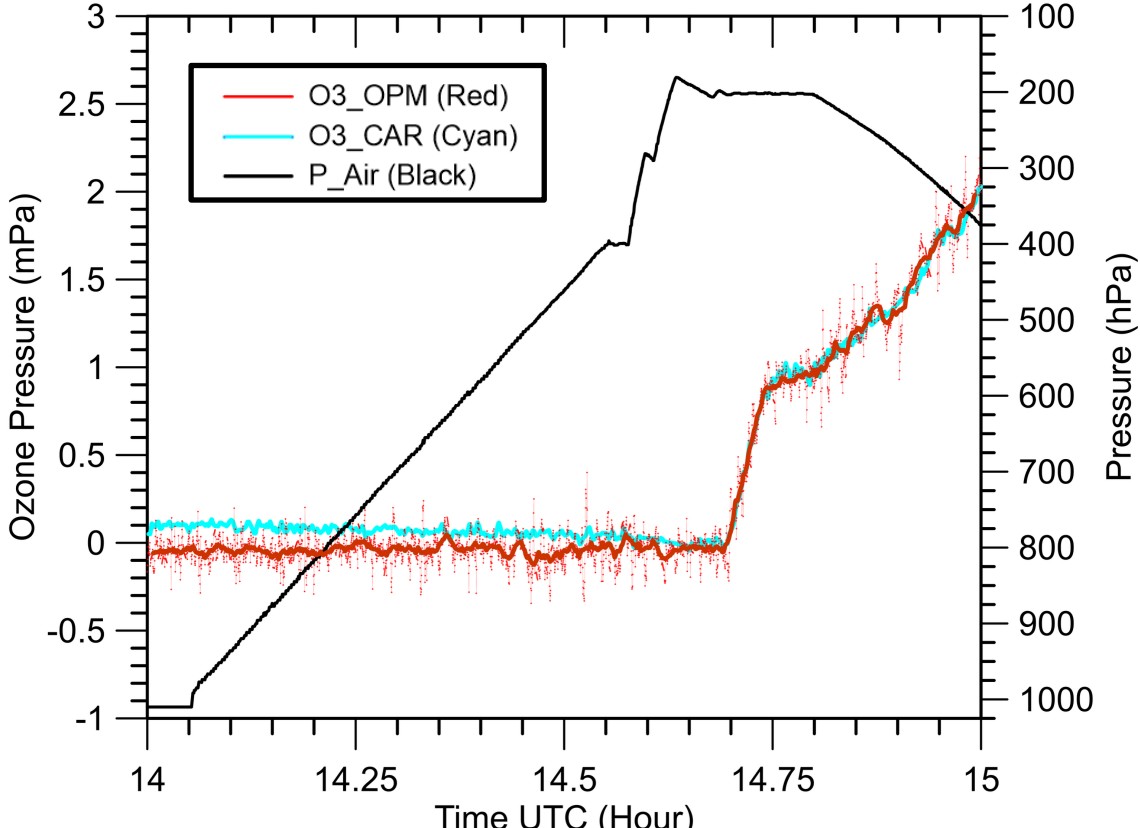

**Figure 12.  Experiment #6:  Time series of pressure (black) and ozone pressure (mPa) for CAR-O3 (cyan=light blue) and OPM (red), while ozone is kept initially at zero and after time = 14.70 (~14:42) hour UTC ozone increased towards 2 mPa.**

## 4. Discussion, Conclusions and Recommendations

In general, the IAGOS-O$_3$ instruments P1-O3 and CAR-O3 as well as the OPM showed consistent and good agreement among each other within a range better than about 5 %. CAR-O3 showed on average about -(1-2) % deviation to the OPM, but no clear pressure dependence within the 1000 hPa down to 400 hPa range, while at 250 hPa CAR-O3 showed about 2-4 % more ozone than the OPM. P1-O3 showed a good performance with a moderate increasing pressure dependent O$_3$ deviation to the OPM of about +2% at 1000 hPa to -3% at 400 hPa. The observed differences are small but systematic. The underlying causes should be better understood, also with respect to how far the observed results are consistent among the suite of instruments flown within IAGOS. Further, an experimental artefact of a few percent cannot be fully excluded, because we had to modify the WCCOS-JOSIE experimental setup to be able to adapt to the large sampling volume flow rate of about 24 lv/min of the P1-O3 (Section 2.3.1). However, no indications are found on any memory or hysteresis effects for both instruments. For IAGOS-O$_3$ the long-term stability of the base line of the measured ozone records is extremely important to derive long term ozone changes of the order of one percent per decade.

Further, the intercomparison experiments here have shown that the reproducibility of the performance of the OPM as a standard, in combination with the experimental set up, within about ±1 %. A primary standard for O3-UV photometer measuring only exists at Earth surface conditions, a primary standard exists at the Bureau International des Poids et Mesures (BIPM), Paris, France, but not for the free atmosphere or at reduced pressure. Therefore, even all intercomparisons in the past like JOSIE (comparison of ozonesondes against OPM) as well as this study (IAGOS-O$_3$ versus OPM) must be interpreted as being relative to each other. Hereby in this intercomparison the OPM acts as the reference instrument. However, it is an important gap in doing intercomparison studies like this that no ozone reference instrument running at reduced pressures exists at any National Metrological Institute in the world. For the global observation networks of measuring free atmospheric ozone concentrations, it is crucial to close this gap to harmonize long term ozone records from different platforms (e.g. aircraft or balloon sondes) to one reference standard.

This intercomparison is a first step with the goal to get the global ozone sonde data (GAW-NDACC-SHADOZ-GRUAN) and IAGOS-O3 (CORE & CARIBIC) data traceable to one common reference (OPM of WCCOS). While the aircraft and sonde measurements are often complementary, their records do not typically cover the same period. It is therefore essential to know and quantify potential biases and characteristics over time when merging their long-term records for process or trend studies. Tarasick et al. (2019) has evaluated earlier in-flight comparisons with ozonesonde measurements within a certain coincidence of space and time (Thouret et al., 1998; Staufer et al., 2013, 2014; Tanimoto et al., 2015) and found a consistent average relative positive bias of 5 % - 10 % between the ozonesondes and IAGOS. In a most recent study (Wang et al., 2024) has confirmed and discussed this observed bias, but no conclusive explanation could be given. It is known that ozone sondes in the troposphere can overestimate ozone by up to 5% (Smit et al., 2007, 2024; Thompson et al., 2019), while aircraft measurements may underestimate ozone due to wall losses when compressing the sampled air before measurement (Dias-Lalcaca et al., 1998; Brunner et al., 2001; Schnadt-Poberaj et al., 2007). However, this intercomparison study has shown that a freshly serviced Pump Box compressing the sampled air to cabin air pressure conditions, before entering the P1-O3 monitor unit of P1-CORE-package, has only a small, in any, impact of less than 2-3% compared to the total measurement error. Further investigations on the performance of the Pump Boxes are needed, particularly the ones which have been flown during long periods of IAGOS-CORE flight operation and thus may have been exposed to highly polluted air masses containing contaminants (e.g. aerosols) near

airports during take-off or landing of the aircraft. A key question thereby is: Can these contaminants have an impact on
the performance of P1-O3 or may the self-cleansing effect through high ozone concentrations, when flying in the
stratosphere, be efficient enough that the impact is small or can be neglected?
A more regular validation of IAGOS-$O_3$ on external consistency is therefore essential and could be achieved by regular
comparisons of the IAGOS-$O_3$ instruments together with ozonesondes against the OPM of the WCCOS in their
environmental simulation chamber. This would be an important milestone in ozone research in the free troposphere and
UTLS.
An important existing gap in doing intercomparison studies like this, however, is that no ozone reference instrument is
running at reduced pressures at any National Metrological Institute in the world. For the global observation networks of
measuring free atmospheric ozone, it is essential to close this gap in the future to enable the traceability of ozone
measurements from different platforms to one reference standard, which is crucial to harmonize long-term ozone records
and the detection of long term-changes in the free atmosphere.
However, a key gap in conducting such intercomparison studies is that no national metrology institute in the world is
hosting and operating a reduced-pressure ozone reference instrument as the common primary standard. It is essential for
global observation networks measuring free atmospheric ozone to close this gap in the future, enabling traceability of
ozone measurements from different platforms to one reference standard. This is crucial for harmonizing long-term ozone
records to detect long-term changes of ozone in the free atmosphere.
**Acknowledgement**
The WCCOS has been sponsored by the Forschungszentrum Jülich GmbH and WMO-GAW. IAGOS is supported by the
European Commission, Airbus and the airlines (Deutsche Lufthansa, Air France, Austrian Airlines, Air Namibia, Cathay
Pacific, Iberia, China Airlines, Hawaiian Airlines, and Air Canada so far) that have carried the MOZAIC or IAGOS
equipment and performed the maintenance since 1994. IAGOS has been funded by the European Union projects IAGOS-
DS and IAGOS-ERI. Additionally, IAGOS has been funded by INSU-CNRS (France), Météo-France, Université Paul
Sabatier (Toulouse, France) ), Forschungszentrum Jülich GmbH and Karlsruher Institut für Technologie. The IAGOS
database is supported in France by AERIS (https://www.aeris-data.fr).

**Code availability:** The software code can be provided on request by Herman G.J. Smit.

**Code and Data availability:** All the data used in this study is available at the IAGOS-data base.
*Note: DOI and https-link are both in preparation and will be available in course of August 2023 to be included in this*
*publication).*

**Interactive computing environment:** N/A.

**Sample availability:** N/A.

**Video supplement:** N/A.

**Supplement link:** *(will be included by Copernicus).*

**Team list:** N/A.

**Author contribution:** (i) the concept of this study was developed and worked out by HGJS, AZ and RB; (ii) data collection
and processing TG, HGJS, FO, RB and JMC; (iii) data analysis by HGJS, FO and RB; (iv) preparation of the manuscript
has been led by HGJS with the support of all co-authors. Data provision by HGJS (OPM and WCCOS data), AZ (IAGOS:
CAR-O3) and RB (IAGOS: P1-O3).

**Disclaimer:** N/A.

**Special issue statement:** Tropospheric Ozone Assessment Report Phase II (TOAR-II) Community (ACP/AMT/BG/GMD
inter-journal SI).
**Review statement:** This paper was edited by Troy Thornberry and reviewed by two anonymous referees.
**Competing interests**
One of the co-authors Andreas Zahn is a member of the editorial board of Atmospheric Measurement Techniques. The
peer review process will be guided by an independent editor. The authors have no other competing interests to declare.
**List of Acronyms**

| 579 | **ASOPOS** | Assessment of Standard Operating Procedures for OzoneSondes |
| 580 | **CARIBIC** | Civil Aircraft for the Regular Investigation of the atmosphere Based on an Instrument Container |
| 581 | **CCQM-GAWG** | Consultative Committee for Amount of Substance: Metrology in Chemistry and Biology- |
| 582 | | Gas Analysis Working Group |
| 583 | **CO** | Carbon monoxide |
| 584 | **DS** | Design Study |
| 585 | **ECC** | Electrochemical Concentration Cell |
| 586 | **ERI** | European Research Infrastructure |
| 587 | **ESC** | Environmental Simulation Chamber |
| 588 | **FTIR** | Fourier Transform Infra-Red spectroscopy |
| 589 | **FZJ** | ForschungsZentrum Jülich |
| 590 | **GAW** | Global Atmosphere Watch |
| 591 | **GCOS** | Global Climate Observing System |
| 592 | **GRUAN** | GCOS Reference Upper Air Network |
| 593 | **HALO** | High Altitude and Long-Range Research Aircraft |
| 594 | **IAGOS** | In-service Aircraft for a Global Observing System |
| 595 | **INSU** | Institut National des Sciences de l'Univers |
| 596 | **IPCC** | Intergovernmental Panel on Climate Change |

| 597 | **JOSIE** | Jülich OzoneSonde Intercomparison Experiment |
|---|---|---|
| 598 | **KIT** | Karlsruher Institut für Technologie |
| 599 | **LIDAR** | Light Detection and Ranging |
| 600 | **MOZAIC** | Measurement of OZone and water vapor by AIrbus in-service airCraft (now IAGOS) |
| 601 | **NDACC** | Network for the Detection of Atmospheric Composition Change |
| 602 | **OPM** | Ozone PhotoMeter instrument (used as UV-reference for ECC-ozonesondes at WCCOS) |
| 603 | **OPS** | Ozone Profile Simulator |
| 604 | **SHADOZ** | Southern Hemisphere ADditional OZonesonde |
| 605 | **SPARC** | Stratosphere-troposphere Processes And their Role in Climate |
| 606 | **STP** | Standard Temperature (=273.15 K) and Pressure (=1013.25 hPa) conditions |
| 607 | **TEI** | Thermo Environmental Instruments |
| 608 | **TOAR** | Tropospheric Ozone Assessment Report |
| 609 | **UNEP** | United Nations Environment Programme |
| 610 | **UTC** | Coordinated Universal Time |
| 611 | **UV** | Ultra-Violet |
| 612 | **VMR** | Volume Mixing Ratio |
| 613 | **WCCOS** | World Calibration Centre for OzoneSonde |
| 614 | **WMO** | World Meteorological Organization |

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
