# Peer review of "Intercomparison of IAGOS-CORE, IAGOS-CARIBIC and"

_EGUsphere, 2024_

## Referee Comment (RC1)

Manuscript Review - Smit et. al, 2025

Journal:  EGUsphere
Date Submitted:  13 February 2025

Title:  Intercomparison of IAGOS-CORE, IAGOS-CARIBIC and WMO/GAW-WCCOS Ozone Instruments at the Environmental Simulation Facility at Julich, German

Authors:  Smit, Galle, Blot, et al.

Summary:

This paper provides a summary of a number of experiments conducted at the Forschungszentrum Julich (FZJ) atmospheric profile simulation chamber designed to connect the ozone measurements profile measurements made as part of the In-service Aircraft for a Global Observation System (IAGOS) to those made on balloon sondes using the common, world-standard UV calibration instrument of Proffitt et al. (1982).  As such this paper is an important contribution to the literature and will allow a harmonization of in situ ozone profiles across these measurement platforms.

In particular, the experiments conducted examined the performance of two versions of the aircraft O3 instruments (P1-O3 and CAR-O3) against the dual-beam UV-Ozone Photometer (OPM) of the World Calibration Center of Ozone Sondes (WCCOS) at FZJ.  The instruments generally showed agreement to within 5-6% over the range of pressures studied.  Interestingly, the P1-O3 instrument showed a consistent trend in offset from the OPM, starting at ~+2% at 1000 hPa and changing linearly to ~ -3% by 400 hPa.  The paper was uncertain as to the cause, which does need to be identified and reconciled.  It mentions that the performance of this instrument might be an artifact of the experimental set-up.  That question should be resolved.

Recommendation:

Publish with relatively minor revisions – see my detailed comments below.

While it would be good to have the question of the drift in the offset of the P1-O3 instrument from the WPM resolved, it is worth getting these results into the literature sooner than later.  If it is not resolved by the time of publication of this manuscript, a follow-up "note" should be submitted with an answer to the question.

Detailed comments:

Line 28:  How often do you recommend that these comparisons be performed? Is it possible that the offsets historically might have looked different if these comparisons had been done before?  How stable are the instruments?

Line 36:  I might say, "anthropogenically influenced"

Line 38:  delete "and its impact on life on Earth."

Line 39:  "Besides the traditional" (change first word and delete "of")

Line 45:  "long-term"

Line 47:  "operation,"

Line 47:  "(9-12 km)" should match later reference of 10-12.5 km cruise altitude (line 78) – one or the other.

Line 47-48:  "and **provides profiles of ozone from the surface to cruise altitude** during take-off and landing**.  S**ince August 1994 more than 70,000…"

Line 53:  swap "instruments" in for "devices"

Line 66:  "chamber,"

Line 68:  "...that is **approved as** EASA…"

Line 78:  "(Z=10-12.5 km)" should match earlier reference of 9-12 km cruise altitude (line 47) - one or the other.

Line 114:  "considered**,** too, **and that** is also dependent…"

Line 118:  "...serves as **the** reference **instrument**."

Line 124:  Delete the phrase, "It is to mentioned that"

Line 137:  What are the flow rates and cell volumes?  I think this question is answered in the text below – just curious about the "flush" time for the cells when they are switched into zero air sampling mode.

Line 148:  swap "evaluate" for "prove"

Line 149: "...is responding linearly to ozone to within…" instead of "...that its linearity is…"

Line 150:  "...comparison with **an instrument with measurements** traceable to the National Institute…"

Line 151:  Confused about the word "above" at the end of this sentence.

Line 152:  "...IAGOS-CORE **is** compared…"

Line 209:  swap "supply equivalent air samples to" in for "provide"

Line 224:  "CAR-O3 uses a 2 m tube (ID = 4 mm), while the P1-O3 inlet line goes…"

Line 242-243:  "...the manifold **was** monitored…"

Line 246:  "...when P1-O3 **was** directly connected…"

Line 251-252:  "...of the aircraft**.  O**n IAGOS-CARIBIC…"

Line 253:  "...about 12.5 km**,** the lowest…"

Line 255:  "...thus to **a value of the** lowest total…"

Line 256:  "Note**,** however, as P1-O3…"

Line 258:  "...both instruments **span** the relevant…"

Line 262:  "...usually **falls in the range** 800-850…"  Question:  By your argument in the text above this point, if you are determining the maximum difference, should you not be using 250 hPa here instead of 280 hPa?  That would result in a 600 hPa difference instead of 570 hPa. Maybe I misunderstood the goal?

Line 268:  "...instrument **does not** use a pump…"

Line 279:  "...simulation experiments**,** number**ed** 3 to 7, which…"

Line 295:  For Figure 2, it might be helpful to include the 0% difference line across the top plot.

Line 304:  "...underlying cause**.  In** a subsequent test (May 2024), KIT found an issue…"  What is "KIT"?

Line 307:  "...after the **repair** of the AD-converter…"

Line 310-311:  "...compared to the OPM.  We only will present the pressure corrected…"

Line 315:  Figure 3 – the panels seems somewhat blurry.  Do you have higher resolution versions for inclusion in the final publication?

Line 320:  "...and the **cruise altitude** section at 400 hPa **(Figure 4b)**."

Line 321:  "...during ascent and descent**,** and no indication…"

Line 354 and elsewhere:  It would be good to formulaically define what you mean by relative difference.

Line 363:  Figure 6 – what are the spikes that appear in the red trace at the step changes in ozone concentration?  Are those real or artifacts of the processing/measurement system?

Line 380:  "**..to within +/- 3%.**"

Line 392:  "...at discrete values **typically found at** the corresponding…"

Line 398: "...again (as in **Exp #3** and #4) **around** -(1-2)% and is constant…"

Line 399: "...400 - 1000 hPa **with** ozone volume mixing ratios…"

Line 399-400:  "...follow the **changes in** ozone levels below 100 ppbv**,** only relative…"

Line 401-402:  "...CAR-O3 in more detail**. Figure 10 shows** the three ozone VMR…"

Line 418-420:  The lines-of-best-fit in Figure 10 are forced through the origin.  Is that the right approach?

Line 420:  "...P1-O3 and CAR-O3" space added between "O3" and "and"

Line 424:  replace "for" with "of"

Line 427-428:  Should the formula be:  "100% X (P1-O3 - OPM)/OPM?  Same for the CAR-O3 calculation…

Line 443:  "**(i.e., the** pressure under real flight conditions, see section 2.3.2)."

Line 446:  "...10 minutes**, the difference** declined to…"

Line 448:  "...the OPS and ESC**;** however, no indication of any **malfunction** of any…"

Line 448-449:  Delete "Although"..."The cause is **still** not understood**;** it is **a** subject for…"

Line 455-456:  "**unrealistically** high and most likely **impacted by the** high temperatures of the electronics of the instrument."  Is there a way to check this hypothesis?

Line 468:  "P1-O3 showed a good performance…"  Should the consistent slope and the 5% change from bottom to top worry users?  Can this be corrected?

Line 473-475:  Comment – if you had swapped the positions of the instruments in the chamber set-up, could you have possibly seen different results?

Line 477-478:  "...of the OPM as a standard, in combination…"

Line 478-479:  "...set up, to within about +/- 1%.  A primary standard for O3-UV photometer measuring only exists at Earth surface conditions (at the Bureau…"

Line 480:  delete "respectively"

Line 483:  "...**reference** instrument."  delete "to refer to"

Line 487:  "...complementary, their records do not typically cover the same **time** period."

Line 497:  "...has only a small, **in any,** impact of less…"

Line 499:  "...**have** been flown…"

Line 502:  "...stratosphere, be efficient **enough** that the impact…"

Line 503:  "...essential, **and** could be…"

Line 538:  LIDAR = "Light Detection and Ranging"

Line 548:  I believe UTC = "Coordinated Universal Time"

---

## Author Comment (AC1)

**Review of Smit et al., *Intercomparison of IAGOS-CORE, IAGOS-CARIBIC and WMO/GAW-WCCOS Ozone Instruments at the Environmental Simulation Facility at Julich, Germany***

*Reply to referee #1*

*We thank referee #1 for the complete and thoughtful review of our manuscript and providing thoughtful comments and suggestions that have helped us improve this manuscript. We also thank Editor Troy Thornberry for handling our paper and coordinating the reviews. Our responses to reviewer comments are provided below in red italic text.*

**Anonymous Referee #1, 18 February 2025**

**Summary:**

This paper provides a summary of a number of experiments conducted at the Forschungszentrum Julich (FZJ) atmospheric profile simulation chamber designed to connect the ozone measurements profile measurements made as part of the In-service Aircraft for a Global Observation System (IAGOS) to those made on balloon sondes using the common, world-standard UV calibration instrument of Proffitt et al. (1982). As such this paper is an important contribution to the literature and will allow a harmonization of in situ ozone profiles across these measurement platforms.

In particular, the experiments conducted examined the performance of two versions of the aircraft O3 instruments (P1-O3 and CAR-O3) against the dual-beam UV-Ozone Photometer (OPM) of the World Calibration Center of Ozone Sondes (WCCOS) at FZJ. The instruments generally showed agreement to within 5-6% over the range of pressures studied. Interestingly, the P1-O3 instrument showed a consistent trend in offset from the OPM, starting at ~+2% at 1000 hPa and changing linearly to ~ -3% by 400 hPa. The paper was uncertain as to the cause, which does need to be identified and reconciled. It mentions that the performance of this instrument might be an artifact of the experimental set-up. That question should be resolved.

**Recommendation:**

Publish with relatively minor revisions – see my detailed comments below.

While it would be good to have the question of the drift in the offset of the P1-O3 instrument from the WPM resolved, it is worth getting these results into the literature sooner than later. If it is not resolved by the time of publication of this manuscript, a follow-up "note" should be submitted with an answer to the question.

*>>> We share the concerns that this study needs as soon as possible more experimental efforts to obtain more statistical evidence concerning the observed drift of the P1-O3 instrument, but also to exclude any artifacts of the experimental set-up. Unfortunately, on the short term it is not possible to do any intercomparison experiments due to a shortage of operational P1-O3 instruments within IAGOS because nowadays all instruments are either "flying" or have been already scheduled to be flown in the upcoming year. However, we think the results of this study should be available to the scientific community the sooner the better. And we fully agree with your recommendation to do follow up intercomparison experiments and that they should be done as soon as possible. At present more intercomparisons are planned for 2026/2027. As soon as we have new results we will publish this of course as a "note".*

**Detailed comments:**

Line 28: How often do you recommend that these comparisons be performed? Is it possible that the offsets historically might have looked different if these comparisons had been done before? How stable are the instruments?

*>>> At present it is too early to make a specific recommendation on how often this kind of intercomparison should be done. We first need more statistics on the variability of the observed differences to characterize these. However, it is not expected that historically the offsets would have been changing over time because the long-term stability of the tested ozone instruments have been regularly checked at laboratory pressure conditions over decades against NIST traceable ozone UV-photometers and no significant change in the offset or slope have been observed. For IAGOS-CORE a suite of ozone UV photometers have been flown since 1994 and all instruments consequently have been checked in the laboratory against a common (MOZAIC) ozone equipment (L147-L156). The long-term stability of the instruments is very good. This is confirmed in a comprehensive study by Blot et al. (AMT, 2021) investigating the internal consistency over more than 25 year of in-flight IAGOS-Core ozone measurements. It was shown that the stability among all the flown P1-O3 instruments is better than within a few percent.*

Line 36: I might say, "anthropogenically influenced"

*>>> Done.*

Line 38: delete "and its impact on life on Earth."

*>>> Done.*

Line 39: "Besides the traditional" (change first word and delete "of")

*>>> Done.*

Line 45: "long-term"

*>>> Done.*

Line 47: "operation,"

*>>> Done.*

Line 47: "(9-12 km)" should match later reference of 10-12.5 km cruise altitude (line 78) – one or the other.

*>>> Altitude ranges have been matched into 10-12.5 km.*

Line 47-48: "and provides profiles of ozone from the surface to cruise altitude during take-off and landing. Since August 1994 more than 70,000…"

*>>> Done.*

Line 53: swap "instruments" in for "devices"

*>>> Done.*

Line 66: "chamber,"

>>> *Done.*

Line 68: "...that is approved as EASA…"

>>> *Done.*

Line 78: "(Z=10-12.5 km)" should match earlier reference of 9-12 km cruise altitude (line 47) - one or the other.

>>> *Done.*

Line 114: "considered, too, and that is also dependent…"

>>> *Done.*

Line 118: "...serves as the reference instrument."

>>> *Done.*

Line 124: Delete the phrase, "It is to mentioned that"

>>> *Done.*

Line 137: What are the flow rates and cell volumes? I think this question is answered in the text below – just curious about the "flush" time for the cells when they are switched into zero air sampling mode.

>>> *Based on cell volume (V_Cell) and volume flow rates (VFR_Cell)) the average residence time of an air sample in a cell: ART_Cell) = V_Cell / VFR_Cell)*

*OPM-O3: V_Cell = 50.6 cm3 & VFR_Cell = 4 l/min: ART_Cell = 0.75 s*

*P1-O3: V_Cell ≈ 30 cm3 & VFR_Cell = 1 l/min: ART_Cell = 1.70 s*

*CAR-O3: V_Cell = 9.5 cm3 & VFR_Cell = 1 l/min : ART_Cell = 0.57 s*

*Critical is that after switching the cells from measuring mode into zero mode or vice versa, the flushing times are long enough such that the air in the cells is refreshed by more than 99.5 %. Experiments in the past have shown that usually after flushing times of about two times the specific ART_Cell this 99.5 % refreshing criterion is fulfilled. This applies for all instruments in discussion, P1-O3, CAR-O3 and OPM-O3 (see alsoTable 1).*

Line 148: swap "evaluate" for "prove"

>>> *Done.*

Line 149: "...is responding linearly to ozone to within…" instead of "...that its linearity is…"

>>> *Done.*

Line 150: "...comparison with an instrument with measurements traceable to the National Institute…"

*>>> Done.*

Line 151: Confused about the word "above" at the end of this sentence.

*>>> The sentence at L151 is corrected into: "The overall uncertainty is better than ± 2 ppbv for $O_3$ < 100 ppbv and ±2 % for $O_3 \geq$ 100 ppbv. (Nédélec et al., 2015)." This has been also corrected in Table 1.*

Line 152: "...IAGOS-CORE is compared…"

*>>> Done*

Line 209: swap "supply equivalent air samples to" in for "provide"

*>>> Done.*

Line 224: "CAR-O3 uses a 2 m tube (ID = 4 mm), while the P1-O3 inlet line goes…"

*>>> Done.*

Line 242-243: "...the manifold was monitored…"

*>>> Done.*

Line 246: "...when P1-O3 was directly connected…"

*>>> Done.*

Line 251-252: "...of the aircraft. On IAGOS-CARIBIC…"

*>>> Done.*

Line 253: "...about 12.5 km, the lowest…"

*>>> Done.*

Line 255: "...thus to a value of the lowest total…"

*>>> Done.*

Line 256: "Note, however, as P1-O3…"

*>>> Done.*

Line 258: "...both instruments span the relevant…"

*>>> Done.*

Line 262: "...usually falls in the range 800-850…" Question: By your argument in the text above this point, if you are determining the maximum difference, should you not be using 250 hPa here instead of 280 hPa? That would result in a 600 hPa difference instead of 570 hPa. Maybe I misunderstood the goal?

*>>> You are right. Thanks for attending us to this mistake. We have corrected the numbers accordingly in the text of the revised manuscript.*

Line 268: "...instrument does not use a pump…"

*>>> Done.*

Line 279: "...simulation experiments, numbered 3 to 7, which…"

*>>> Done.*

Line 295: For Figure 2, it might be helpful to include the 0% difference line across the top plot.

*>>> Included. The figures have been revised, majorly on request of referee#2*

Line 304: "...underlying cause. In a subsequent test (May 2024), KIT found an issue…" What is "KIT"?

*>>> KIT = Karslruher Institut für Technologie. Has been included in the text and the List of Acronyms*

Line 307: "...after the repair of the AD-converter…"

*>>> Done.*

Line 310-311: "...compared to the OPM. We only will present the pressure corrected…"

*>>> Done.*

Line 315: Figure 3 – the panels seem somewhat blurry. Do you have higher resolution versions for inclusion in the final publication?

*>>> The figure has been exchanged by a higher resolution one.*

Line 320: "...and the cruise altitude section at 400 hPa (Figure 4b)."

*>>> Done.*

Line 321: "...during ascent and descent, and no indication…"

*>>> Done.*

Line 354 and elsewhere: It would be good to formulaically define what you mean by relative difference.

*>>> We add at L292 the exact definitions of the relative differences:*

*The relative differences in % of the $\mu_{O3}$ (VMR) readings of P1-O3 and CAR-O3, respectively, shown in this study are consequently defined with regard to the $\mu_{O3}$ readings of the OPM-O3 instrument acting as the reference as follows:*

$$Rel.\,Difference\,of\,P1O3 = \frac{(\mu_{O3,P1O3} - \mu_{O3,OPMO3})}{\mu_{O3,OPMO3}} \qquad (3)$$

$$Rel.\,Difference\,of\,CARO3 = \frac{(\mu_{O3,CARO3} - \mu_{O3,OPMO3})}{\mu_{O3,OPMO3}} \qquad (4)$$

Line 363: Figure 6 – what are the spikes that appear in the red trace at the step changes in ozone concentration? Are those real or artifacts of the processing/measurement system?

*>>> The $O_3$ spikes observed at the step changes are real and are caused by the "overshooting effect" of the fast response of the flow controllers of the OPS (Ozone Profile Simulator) on the sudden, abrupt change of the settings of the simulated $\mu_{O3}$ profile (see also section 2.2.2 on the OPS).*

Line 380: "..to within +/- 3%."

*>>> Done.*

Line 392: "...at discrete values typically found at the corresponding…"

*>>> Done.*

Line 398: "...again (as in Exp #3 and #4) around -(1-2)% and is constant…"

*>>> Done.*

Line 399: "...400 - 1000 hPa with ozone volume mixing ratios…"

*>>> Done.*

Line 399-400: "...follow the changes in ozone levels below 100 ppbv, only relative…"

*>>> Done.*

Line 401-402: "...CAR-O3 in more detail. Figure 10 shows the three ozone VMR…"

*>>> Done*

Line 418-420: The lines-of-best-fit in Figure 10 are forced through the origin. Is that the right approach?

*>>> We have changed the approach of Fig.10 and Table 3. As before the offsets of the three instruments have been determined during the periods of zero ozone exposure, while the slopes were obtained from linear fits of the scatter plots in Fig.10-a., b., c. respectively but not forced through the origin. In addition, similar graphs are shown for the lower ozone ranges for the corresponding three pressure levels (Fig.10-d., e., f.). For the 6 graphs in Fig.10 we also determined the slopes for the periods of upward and downward ozone step levels. The corresponding scatter plots and linear fits are displayed in Fig. S1 and S2 of the Supplement for the P1-O3/ OPM and CAR-O3/OPM, respectively. All results are summarized in Table 3 and discussed in the new manuscript in following paragraph:*

*"From Table 3 it is seen that the behaviour between the three instruments observed at ozone levels larger than about 100 ppbv is consistent with the results obtained from the Exp. #3 and Exp. #4. At lower ozone values below 100 ppbv, however, the slopes for P1-O3/OPM differ slightly by -(1-2) % compared to their corresponding slopes of P1-O3/OPM derived for higher ozone values, respectively. Breaking down the slopes into the upward and downward part of the ozone step levels, P1-O3/OPM reveals a small hysteresis effect of about a 2 % which is most pronounced in the lower range of ozone levels. CAR-O3 shows no hysteresis, neither at the higher nor at the lower ozone levels (Table 3 and Figs. S1 and S2 in the supplement). The observed differences are not really understood but are still within the experimental reproducibility of about ±1 % as mentioned in Section 3.2.2. "*

Line 420: "...P1-O3 and CAR-O3" space added between "O3" and "and"

*>>> Done.*

Line 424: replace "for" with "of"

*>>> Done.*

Line 427-428: Should the formula be: "100% X (P1-O3 - OPM)/OPM? Same for the CAR-O3 calculation…

*>>> We add at L292 the exact definitions of the relative differences where it explicitly mentioned that these definitions (Eq.3) and (Eq.4) throughout this entire study consequently will be used.*

Line 443: "(i.e., the pressure under real flight conditions, see section 2.3.2)."

*>>> Done.*

Line 446: "...10 minutes, the difference declined to…"

*>>> Done.*

Line 448: "...the OPS and ESC; however, no indication of any malfunction of any…"

*>>> Done.*

Line 448-449: Delete "Although"..."The cause is still not understood; it is a subject for…"

*>>> Done.*

Line 455-456: "unrealistically high and most likely impacted by the high temperatures of the electronics of the instrument." Is there a way to check this hypothesis?

*>>> From our long year JOSIE experiences we have observed that when the temperature of the lamp or the light detector electronics are exceeding 50 $^{o}$C, the detector signal noise is getting larger and increasing with increasing temperatures. Lamp or light detector temperatures larger than 45-50 $^{o}$C are observed only incidentally after long operation times and which most likely occurs when the laboratory temperature is above 25-30 $^{o}$C. This was the case at the end of that day of doing the experiment. We changed the sentence accordingly:*

*"The OPM showed a small negative offset about - (0.05 – 0.10 mPa), but a rather noisy signal, unrealistic high and most likely due to the enhanced temperatures of the UV-light detector electronics exceeding the 50 °C threshold that occured during the experiment."*

Line 468: "P1-O3 showed a good performance…" Should the consistent slope and the 5% change from bottom to top worry users? Can this be corrected?

*>>> Although the observed differences among the three different instruments are in general consistent within about a 5 % range, the P1-O3 instrument reveals a small but consistent pressure dependent change of about 5 % compared to the OPM. the results are significant to alert the community, but further investigations in the near future are needed to analyze P1-O3 versus OPM characteristics. At present, it is too early to draw solid conclusions to recommend any corrections of the P1-O3 data because this study deals only with the results of a first intercomparison of one single P1-O3 instrument against the OPM of the WCCOS. More intercomparisons of several different P1-O3 instruments versus OPM are needed.*

Line 473-475: Comment – if you had swapped the positions of the instruments in the chamber set-up, could you have possibly seen different results?

*>>> We did not really swapped the positions of the instruments, but at the beginning of the campaign we changed the connection of the P1-O3 to the manifold as written in section 2.3.1 (Experimental Setup at L244-L247 in the original manuscript) : "The P1-O3 sample flow we had to branch off from the ozone profile simulator flow before entering the manifold (Fig.1), because it was shown that the high sampling volume rate of P1-O3 pump box would otherwise cause leakage effects when P1-O3 was directly connected with a Teflon fitting at the inlet glas tube of the manifold." The leakage effects were causing P1-O3 readings with a pressure dependent deviation of -5% at lab pressure up to -15% at lower pressure compared to the OPM. After we branched off the P1-O3 inlet tube from the ozone profile simulator flow before entering the manifold (See Fig.1) the effect seemed to be gone, however, a remaining pressure dependent artefact cannot be fully excluded (See also L471-L473 in the original manuscript).*

*>>> By simply swapping the positions of the instruments connected to the ozone manifold we didn't expect to obtain different results because the air pressure $P_M$ inside the manifold is strictly kept a few hPa higher than the air pressure inside the chamber, such that any leakage of chamber air into the manifold would be avoided (L242-L244 in the original manuscript).*

Line 477-478: "...of the OPM as a standard, in combination…"

*>>> Done.*

Line 478-479: "...set up, to within about +/- 1%. A primary standard for O3-UV photometer measuring only exists at Earth surface conditions (at the Bureau…"

*>>> Done.*

Line 480: delete "respectively"

*>>> Done.*

Line 483: "...reference instrument." delete "to refer to"

*>>> Done.*

Line 487: "...complementary, their records do not typically cover the same time period."

*>>> Done.*

Line 497: "...has only a small, in any, impact of less…"

*>>> Done.*

Line 499: "...have been flown…"

*>>> Done.*

Line 502: "...stratosphere, be efficient enough that the impact…"

*>>> Done.*

Line 503: "...essential, and could be…"

*>>> Done*

Line 538: LIDAR = "Light Detection and Ranging"

*>>> Done.*

Line 548: I believe UTC = "Coordinated Universal Time"

*>>> Done.*

---

## Author Comment (AC2)

**Review of Smit et al., *Intercomparison of IAGOS-CORE, IAGOS-CARIBIC and WMO/GAW-WCCOS Ozone Instruments at the Environmental Simulation Facility at Julich, Germany***

*Reply to referee #2*

*We thank referee #2 for the complete and thoughtful review of our manuscript and providing thoughtful comments and suggestions that have helped us improve this manuscript. We also thank Editor Troy Thornberry for handling our paper and coordinating the reviews. Our responses to reviewer comments are provided below in red italic text.*

**Anonymous Referee #2, 08 May 2025**

As mentioned, several times in the text this intercomparison study is a first step of the long-term goal to get the global ozone sonde data and IAGOS-O3 data sets traceable to one common reference. This long-term goal is extremely important for the ozone research community and therefore this study is very welcome! Accordingly, I recommend publication after addressing two major and the minor comments below.

**Major**

1. As mentioned in the text twice there is no ozone reference instrument running at reduced pressures at any National Metrological Institute in the world. The OPM at Julich acts as a workaround to this situation. However, we need such a reference in addition to studies like this one. Although this gap in the whole concept is mentioned in the manuscript, I recommend to address it even more prominent and pronounced eg. in the abstract. This would enable relevant persons better to ask for closing this gap.

*>>> Thanks for the suggestion and recommendation.*

*We added an extra sentence at the end of the abstract and a similar sentence at the end of the second paragraph of Chapter 4:*

*"An important existing gap in doing intercomparison studies like this is that no ozone reference instrument is running at reduced pressures at any National Metrological Institute in the world. For the global observation networks of measuring free atmospheric ozone, it is essential to close this gap to enable the traceability of ozone measurements from different platforms to one standard reference, which is crucial to harmonize long-term ozone records to detect long term-changes of ozone in the free atmosphere."*

2. The display and description of the figures can be crucially improved. Each single issue is a minor one (and are listed in the minor section below). However, overall, these issues make it sometimes hard to follow the arguments of the manuscript.

*>>> Thank you very much for the many suggestions you did to revise the figures. Thereby, it has been attempted that all figures are more harmonized by following same standards. The changes are a substantial improvement to read and understand the figures better and more easily.*

**Minor and typos**

Figure 2:

There is no temperature data and the y-axis "Temperature" text can be omitted. Please arrange the figure similar to Figs. 4b, 6, 7b, 9, and 11, ie. separate upper and lower panel. Here and in other figures, please use lower case letters for ppbv. There is no need to write VMR in addition to "Volume Mixing Ratio" at the y-axis. For this type of figure, it would be nice to use both sides of the frame for y-axes descriptions instead of an additional y-axis on the left side.

>>> *The figures have been revised following your suggestions.*

Figure 3:

Some blue dots are masked by color descriptions. Please avoid that. Less cryptic axes descriptions would be welcome, eg. "reference pressure" instead of "p-ref.", "pressure difference" instead of "Delta p". At least explain abbreviations if used in the plots, eg. "Delta p", "cuv.", "ref.", in the caption.

>>> *Exchange by a new figure of higher resolution.*

Figure 4 (& 7):

Panel letter descriptions, a b c, should be inside the plots. 4a and 4c: Separation into two parts like in 4b with help lines at -5, 0, and 5% would be helpful.

>>> *The figure has been changed following your suggestions.*

Line 38: Please add a comma "... of the atmosphere, and its impact ..."

>>> *The last part of the sentence has been omitted on suggestion by referee#1*

Line 69: The information in the two brackets can be combined.

>>> *Done.*

Line 148: "... to prove that the linearity of the instrument is within 1%."

>>> *The sentence has been slightly revised following your suggestion and that from referee#1*

Line 151: Unsure, what the word "above" should tell.

>>> *Revised the sentence for better understanding.*

Line 152: Is the "s" in "P1s" correct?

>>> *Changed by "Each flown P1-Package"*

Line 287-292: In the description of what can be seen in this figure, please add a comment on missing data, ie. absence of CAR-O3 measurements during about 30 min, partly during descent. Any additional comment, what had happened at 14:40 UTC?

>>> *Incidentally the temperature controller of the UV-LED was not working properly, thus the measured O3 values were flagged as potentially erratic and removed from the final data. We add an extra sentence at L292: "The three missing data intervals of the CAR-O3 instrument were caused by a malfunction of its temperature controller for the UV-LED light source such that the measured O3 values were rejected, not shown in the graph and excluded from further analysis".*

Line 304: My understanding of the following text is, that the issue found with the electronic analog-digital converter was systematic present in all CAR-O3 instruments and not only in the instrument used in this study. Please mention that explicitly here.

*>>> An extra sentence to explain the issue in more detail is included at end of the paragraph at L312. "For the two similar CAR-O3 type instruments (FAIRO-1 and FAIRO-2) flown primarily on the German research aircraft HALO (HALO (High Altitude and Long-Range Research Aircraft), the ADC modules were investigated and found to be configured correctly, such that no correction is needed".*

Line 320-321: The statement "... occurs identically during ascent and descent and no indication for any hysteresis effects could be observe" is hard to verify with the current set of figures. I miss an additional corresponding figure overlaying ascent and descent branches with eg. different colors.

*>>> We fully agree. We dropped here the statement of the hysteresis, but investigated the behaviour of P1-O3 and CAR-O3 relative to the OPM in more detail in Fig. 10 by breaking down the slopes of the scatter fits for the corresponding instruments and pressure levels into the upward and downward parts of the ozone step levels, once for the full ozone range and once for the lower ozone range. The corresponding scatter graphs are shown in Fig. S1 and S2 in the Supplement, and the results of the slopes of the linear fits are summarized in Table 3.*

Line 420-424: Please swap the sequence of both sentences.

*>>> Done*

Line 421-422: The relative differences for larger parts, eg. zero ozone periods, are not shown in the upper panel of figure 9. Why?

*>>> At lower ozone concentrations even small differences between the instruments can easily produce large relative differences. We revised the sentence into:*

*"Here only relative differences for the higher ozone levels are shown in Fig. 9 to avoid that at lower ozone concentrations even small differences between the instruments can easily produce large relative values."*

Line 421/422 & 431/432: The figure numbering seems to be odd. The referenced figure 8 deals with experiment 4 and not with experiment 7. A corresponding figure for experiment 7 is missing. Please add such a figure. Instead figure 9 figure 10 should be referenced.

*>>> The figure number referencing has been corrected: Fig.9 for offsets and Fig.10 for the slopes at the three different pressure levels (950, 600 and 400 hPa). Because we only have results at three discrete pressure levels we dispensed for an additional figure such as Fig.8 but to summarize in Table 3 the results of the derived offsets and slopes at the three different pressure levels and low and high ozone values. The obtained results are at ozone values larger in agreement with the previous experiments #3 (Fig. 5) and #4 (Fig. 8): see also the paragraph L426-429 in the reviewed manuscript (Jan.2025)*

Line 427: Small differences of -(1-2) % for the P1-O3/OPM are mentioned. I see values between -7 % and +3 %. Have I misunderstood something?

*>>> There is indeed a misunderstanding - to avoid this, the paragraph (L426-429) has been rewritten. Please also see our reply on your next comment on Table 3.*

*"Table 3 shows that the behaviour of the three instruments observed at ozone levels larger than about 100 ppbv is consistent with the results obtained from the Exp. #3 and Exp. #4. At lower ozone values below 100 ppbv, however, the slopes for P1-O3/OPM differ slightly by -(1-2) % compared to their corresponding slopes of P1-O3/OPM derived for higher ozone values, respectively. Breaking down the slopes into the upward and downward part of the ozone step levels, P1-O3/OPM reveals a small hysteresis effect of about a 2 % which is most pronounced in the lower range of ozone levels. CAR-O3 shows no hysteresis, neither at the higher nor at the lower ozone levels (Table 3 and Figs. S1 and S2 in the supplement). The observed differences are of unknown origin but still within the experimental reproducibility of about ±1 % as mentioned in Section 3.2.2."*

Table 3: In figure 10 one can see only the fits per instrument for the upper O3-ranges but not for the lower O3-ranges. Another figure filling that gap would be helpful to verify such values like 0.942 for the P1-O3/OPM at 400 hPa / 0-200 ppbv.

*>>> Thanks for the suggestion, we added to the figure also graphs for the lower ozone ranges.*

*>>> In addition, we have changed the approach of Fig.10 and Table 3. As before the offsets of the three instruments have been determined during the periods of zero ozone exposure, while the slopes were obtained from linear fits of the scatter plots in Fig.10-a., b., c. respectively but not forced through the origin. In addition, similar graphs are shown for the lower ozone ranges for the corresponding three pressure levels (Fig.10-d., e., f.). For the 6 graphs in Fig.10 we also determined the slopes for the periods of upward and downward ozone step levels. The corresponding scatter plots and linear fits are displayed in Fig. S1 and S2 of the Supplement for the P1-O3/ OPM and CAR-O3/OPM, respectively. All results are summarized in Table and discussed in the new manuscript in paragraph:*

*"From Table 3 the behaviour between the three instruments observed at ozone levels larger than about 100 ppbv is consistent with the results obtained from the Exp. #3 and Exp. #4. At lower ozone values below 100 ppbv, however, the corresponding slopes for P1-O3/OPM differ slightly by -(1-2) % compared to their corresponding slopes derived at higher ozone values, respectively. Breaking down the slopes into the upward and downward part of the ozone step levels it show that P1-O3 reveals a small hysteresis effect of about a 2 % which is most pronounced in the lower range of ozone levels, while CAR-O3 shows no hysteresis, neither in the higher nor in the lower ozone levels (Table 3 and Figs. S1 and S2 in the supplement). The observed differences are not really understood but are still within the experimental reproducibility of about ±1 % as mentioned in Section 3.2.2. "*

Line 448: malfunction

*>>> Done.*

List of Acronyms: Please add RAM.

*>>> RAM is not an acronym but ram air pressure is only another expression for dynamic air pressure. In L251 we revised the sentence by*

*"……thus use the dynamic (ram) air pressure generated by the high speed of the aircraft……".*

Line 583: Staehelin

*>>> Done.*

General: It would be nice to have somewhere an exact definition of the relative difference used, eg. = (O3_IAGOS - O3_OPM) / O3_OPM

*>>> We add at L292 the exact definitions of the relative differences:*

*The relative differences in % of the $\mu_{O3}$ (VMR) readings of P1-O3 and CAR-O3, respectively, shown in this study are consequently defined regarding the $\mu_{O3}$ readings of the OPM-O3 instrument acting as the reference as follows:*

$$Rel.\,Difference\,of\,P1O3 = \frac{(\mu_{O3,P1O3}-\mu_{O3,OPMO3})}{\mu_{O3,OPMO3}} \qquad (3)$$

$$Rel.\,Difference\,of\,CARO3 = \frac{(\mu_{O3,CARO3}-\mu_{O3,OPMO3})}{\mu_{O3,OPMO3}} \qquad (4)$$